# DisTaC: Conditioning Task Vectors via Distillation for Robust Model Merging

**Kotaro Yoshida**[1][*]   **Yuji Naraki**[2]   **Takafumi Horie**[3]
**Ryotaro Shimizu**[4]   **Ioannis Mitliagkas**[5,6]   **Hiroki Naganuma**[5,6][†]
[1]Institute of Science Tokyo   [2]Independent Researcher   [3]Kyoto University
[4]ZOZO Research   [5]Mila   [6]Université de Montréal

## Abstract

Model merging has emerged as an efficient and flexible paradigm for multi-task learning, with numerous methods being proposed in recent years. However, these state-of-the-art techniques are typically evaluated on benchmark suites that are highly favorable to model merging, and their robustness in more realistic settings remains largely unexplored. In this work, we first investigate the vulnerabilities of model-merging methods and pinpoint the source-model characteristics that critically underlie them. Specifically, we identify two factors that are particularly harmful to the merging process: (1) disparities in task vector norms, and (2) the low confidence of the source models. To address this issue, we propose **DisTaC** (**Dis**tillation for **Ta**sk vector **C**onditioning), a novel method that pre-conditions these problematic task vectors before the merge. DisTaC leverages knowledge distillation to adjust a task vector's norm and increase source-model confidence while preserving its essential task-specific knowledge. Our extensive experiments demonstrate that by pre-conditioning task vectors with DisTaC, state-of-the-art merging techniques can successfully integrate models that exhibit these harmful traits, where they would otherwise fail, and achieve significant performance gains. The source code is available at https://github.com/katoro8989/DisTaC

## 1 Introduction

The recent wave of open-sourcing both large pretrained models (Devlin et al., 2019; Rombach et al., 2022; Achiam et al., 2023; Grattafiori et al., 2024) and their fine-tuned downstream variants (Wolf et al., 2019; Taori et al., 2023) has put an unprecedented variety of neural networks within easy reach of anyone. This democratization has, in turn, accelerated research on model merging (Wortsman et al., 2022b;a; Ilharco et al., 2023; Yadav et al., 2023; Akiba et al., 2025), techniques that create new, customized models by integrating existing fine-tuned models without the need for additional large-scale training. In particular, a flurry of recent methods aims to build multi-task models by merging networks that have been fine-tuned independently for each task, rather than retraining a single shared model from scratch (Ilharco et al., 2023; Yadav et al., 2023; Ortiz-Jimenez et al., 2023; Wang et al., 2024; Yoshida et al., 2025; Gargiulo et al., 2025). Many of these techniques require only minimal extra training or none at all. Compared with conventional multi-task learning (MTL), they offer two key advantages: (i) they eliminate the need to aggregate all task-specific labeled data in one location, sidestepping data-access constraints, and (ii) they make it easy to add or edit the model's skill on a particular task after deployment (Yang et al., 2024a).

On established benchmarks, these approaches have shown promising gains, in some cases approaching the performance of traditional MTL (Gargiulo et al., 2025). Yet those benchmarks are built under conditions that are highly idealized for model merging; how robust current merging methods remain in more practical, pessimistic settings is still largely unknown. Bridging this gap is a prerequisite for real-world application.

---

[*]Corresponding author yoshida.k.0253@m.isct.ac.jp
[†]Corresponding author naganuma.hiroki@mila.quebec.

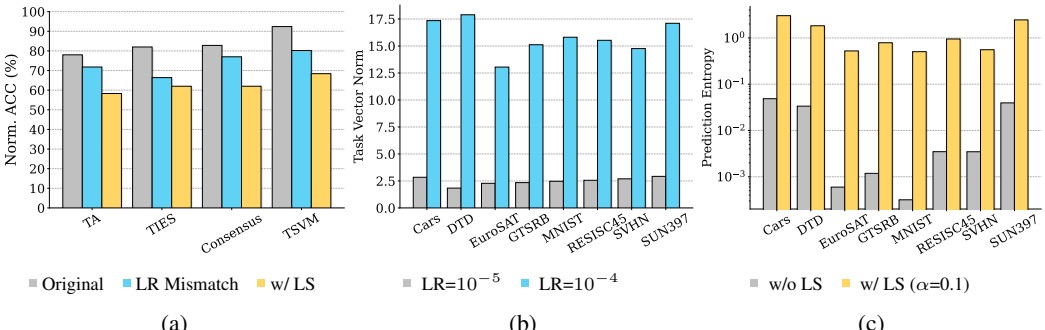

Figure 1: **Failure Cases of Multi-Task Model Merging.** All results were obtained using CLIP with a ViT-B-32 backbone on the eight vision tasks. (a) Comparison of normalized accuracy after merging models from different fine-tuning configurations averaged over eight vision tasks. The gray bar represents the conventional setting (a uniform learning rate of $10^{-5}$ with hard labels). The blue bar indicates the result of merging after training just one task with a learning rate (LR) of $10^{-4}$. The yellow bar shows the result when all tasks were trained with label smoothing (LS). Both the blue and yellow configurations show a significant performance degradation compared to the conventional setting. (b) Change in the task vector norm after fine-tuning with different learning rates for the same number of steps across eight vision tasks. The gray bar uses a learning rate of $10^{-5}$, matching the conventional benchmark, while the blue bar uses $10^{-4}$. We observe a 5 to 7-fold difference in the resulting task vector norms. (c) Change in the entropy of the model's predictive probabilities after fine-tuning with or without label smoothing across eight vision tasks. The vertical axis is on a logarithmic scale. Training with label smoothing increases the entropy by three orders of magnitude.

To that end, we first pinpointed where generic multi-task model merging pipelines break down. Our analysis reveals two especially harmful factors: (1) differences in task vector norms and (2) low prediction confidence of source models. Figure 1a illustrates the vulnerability of recent merging methods to these factors using CLIP (Radford et al., 2021) with a ViT-B-32 (Dosovitskiy et al., 2021) backbone on the eight vision tasks defined in Section 5.1: blue bars show the effect of training models with different learning rates, thereby altering task vector norms (see Figure 1b), while yellow bars show the effect of label smoothing (Müller et al., 2019) (LS), which reduces model confidence (see Figure 1c). In the plot, the horizontal axis lists the merging methods, and the vertical axis reports the average normalized accuracy (Norm. ACC) across the eight tasks, defined as the post-merge accuracy relative to the pre-merge accuracy obtained by individual models for each task. In both cases, every method's performance degrades substantially compared to the standard baseline, represented by the gray bars (a uniform learning rate of $10^{-5}$ with hard labels), with a maximum 24% drop in Norm. ACC.

These failure modes often arise in real-world deployments. For instance, differences in task vector norms can stem from varied learning rates, fine-tuning steps, or weight decay used during the individual fine-tuning of each task (Devlin et al., 2019; Wightman et al., 2021). Low confidence often results from techniques such as LS, Mixup (Zhang et al., 2017), and focal loss (Lin et al., 2017). We therefore contend that models should be pre-conditioned before merging to remove their latent harmfulness. To this end, we propose **Dis**tillation for **Ta**sk-vector **C**onditioning (**DisTaC**) a lightweight knowledge distillation (KD) procedure that tackles both issues using only unlabeled data: To correct task vector–norm disparities, DisTaC first rescales each vector to a chosen target norm and then restores any performance lost through this scaling by distilling knowledge from the original model. To address low source-model confidence, it trains the student with a higher temperature than the teacher ($T_{\text{stu}} > T_{\text{tcr}}$), so the student ultimately produces lower-entropy predictions, that is, predictions that are more confident.

Algorithm 1 combines these two conditioning steps, allowing them to be carried out in a single pass. Because DisTaC leverages the already-trained task vectors as the initialization for KD and relies solely on unlabeled data, it incurs minimal computational overhead and imposes only modest practical requirements, yet markedly improves the robustness of existing model merging techniques in challenging scenarios.

Empirically, on eight vision tasks with ViT-B-32/L-14 backbones, DisTaC increased post-merge accuracy by up to 20.8 percentage points and restored the best-performing TSVM merge's normalized

accuracy from 68% to 92% under low-confidence conditions, thereby matching the conventional "ideal" benchmark performance (i.e., merging high-confidence models with uniform task vector norms), all with minimal computational cost. **Our contributions are as follows:**

- We identify two failure modes in model merging: (i) the task vector norms of the source models differ (Section 3.1), and (ii) the source models' outputs are low-confidence or even well-calibrated (i.e., their predicted probabilities match the true frequency of correctness) (Section 3.2). We provide theoretical explanations and empirical results for each of these phenomena.

- We propose **DisTaC**, a distillation method of source model's weights under appropriate conditions (Section 4), and demonstrate that it mitigates aforementioned failure modes (Section 5.2.1). Our DisTaC is a computationally efficient method, as it requires only a small number of training steps and relies solely on unlabeled data (Section 5.2.2).

- From our analysis, we present two guidelines for model merging: (i) when the task vector norms differ, it is better to shrink the larger vector rather than stretch the smaller one (Section 6.1); and (ii) when the source models have low confidence, it is more effective to make them overconfident before merging, and then apply a calibration method to the merged model (Section 6.2).

## 2 PRELIMINARIES

**Notation.** Let $f(\cdot\,;\boldsymbol{\theta}) : \mathcal{X} \to \mathbb{R}^C$ be a neural network for a $C$-class classification task, parameterized by a vector $\boldsymbol{\theta} \in \mathbb{R}^d$. The network maps an input vector $\boldsymbol{x} \in \mathcal{X} \subseteq \mathbb{R}^D$ to a $C$-dimensional logit vector. We target a multi-task scenario comprising $T$ supervised tasks. Let $\boldsymbol{\theta}_{\mathrm{pre}} \in \mathbb{R}^d$ be the parameters of an open-source pretrained backbone. For each task $t \in \{1, \ldots, T\}$, we obtain a model that has already been fine-tuned on the corresponding labeled dataset $\mathcal{D}_t = \left\{ (\boldsymbol{x}_{t,i}, \boldsymbol{y}_{t,i}) \right\}_{i=1}^{n_t}$, yielding task-specific weights $\boldsymbol{\theta}_t \in \mathbb{R}^d$. Each label $\boldsymbol{y}_{t,i} \in \{0, 1\}^C$ is a one-hot vector indicating the ground-truth class.

### 2.1 MODEL MERGING FOR MULTI-TASK LEARNING

Recent model merging techniques operate on the task vectors (Ilharco et al., 2023) $\boldsymbol{\tau}_t := \boldsymbol{\theta}_t - \boldsymbol{\theta}_{\mathrm{pre}}$ and obtain a single multi–task model by linearly combining them:

$$\boldsymbol{\theta}_{\mathrm{mtl}} = \boldsymbol{\theta}_{\mathrm{pre}} + \sum_{t=1}^{T} \boldsymbol{P}_t \, \boldsymbol{\tau}_t, \tag{1}$$

where each $\boldsymbol{P}_t \in \mathbb{R}^{d \times d}$ is a method-specific matrix that mitigates inter-task interference.

In the following, we explain the $\boldsymbol{P}_t$ used in each merging method.

**Uniform averaging**: $\boldsymbol{P}_t = \frac{1}{T} \boldsymbol{I}_d$.

**Task arithmetic** (Ilharco et al., 2023): $\boldsymbol{P}_t = \lambda_t \boldsymbol{I}_d$, where $\lambda_t \in \mathbb{R}$.

**Ties-Merging** (Yadav et al., 2023): $\boldsymbol{P}_t = \lambda_t \, \boldsymbol{m}_{\mathrm{Ties},t} \, \boldsymbol{I}_d$, where $\lambda_t \in \mathbb{R}$, $\boldsymbol{m}_{\mathrm{Ties},t} \in \{0, 1\}^d$. $\boldsymbol{m}_{\mathrm{Ties},t}$ is determined by the norm of each weight parameter to mitigate inter-task conflicts.

**Consensus Merging** (Wang et al., 2024): $\boldsymbol{P}_t = \lambda_t \, \boldsymbol{m}_{\mathrm{Cons},t} \, \boldsymbol{I}_d$, where $\lambda_t \in \mathbb{R}$, $\boldsymbol{m}_{\mathrm{Cons},t} \in \{0, 1\}^d$. The framework is the same as Ties-Merging, but the binary mask $\boldsymbol{m}_{\mathrm{Cons},t}$ is determined in the following steps. First, create the TALL mask $\boldsymbol{m}_{\mathrm{TALL},t}$, which is a binary mask of weights where each element is set to 1 if the norm of $\boldsymbol{\tau}_t$ is larger than the weighted distance between $\boldsymbol{\tau}_t$ and $\sum_{t=1}^{T} \boldsymbol{\tau}_t$. Then, create $\boldsymbol{m}_{\mathrm{Cons},t}$, where each element is set to 1 if the corresponding element of $\boldsymbol{m}_{\mathrm{TALL},t}$ is 1 in at least $k$ tasks, reflecting agreement among the source models regarding the importance.

**TSVM** (Gargiulo et al., 2025) cannot be expressed within the framework of Eq. (1). Instead, it suppresses task interference by whitening the matrices $\mathbf{U}_t$ and $\mathbf{V}_t$ obtained from the singular value decomposition of the task vectors $\boldsymbol{\tau}_t = \mathbf{U}_t \boldsymbol{\Sigma}_t \mathbf{V}_t^{\top}$.

## 2.2 KNOWLEDGE DISTILLATION

Knowledge distillation (KD) is a model compression and transfer paradigm in which a compact student network is trained to replicate the behavior of a larger, well-performing teacher network (Hinton et al., 2015). By minimizing a joint loss that combines ground-truth supervision with a soft-target signal derived from the teacher's output distribution, the student acquires the teacher's dark knowledge, namely, fine-grained inter-class relationships encoded in the soft logits, while retaining a substantially smaller parameter footprint. Formally, for a given input $\boldsymbol{x}$, let $\boldsymbol{z}_{\mathrm{tcr}} := f(\boldsymbol{x}\,;\boldsymbol{\theta}_{\mathrm{tcr}}) \in \mathbb{R}^C$ and $\boldsymbol{z}_{\mathrm{stu}} := f(\boldsymbol{x}\,;\boldsymbol{\theta}_{\mathrm{stu}}) \in \mathbb{R}^C$ be the output logits from the teacher and student models, parameterized by $\boldsymbol{\theta}_{\mathrm{tcr}} \in \mathbb{R}^d$ and $\boldsymbol{\theta}_{\mathrm{stu}} \in \mathbb{R}^d$, respectively. The KD objective then augments the conventional cross-entropy loss $\mathcal{L}_{\mathrm{CE}}$ with a softened Kullback-Leibler (KL) divergence term:

$$\mathcal{L}_{\mathrm{KD}} = (1-\zeta)\,\mathcal{L}_{\mathrm{CE}}\big(\boldsymbol{z}_{\mathrm{stu}},\,\boldsymbol{y}\big) \ + \ \zeta\,T_{\mathrm{tcr}}T_{\mathrm{stu}}\,\mathrm{KL}\Big(\sigma(\boldsymbol{z}_{\mathrm{tcr}}/T_{\mathrm{tcr}}) \,\big\|\, \sigma(\boldsymbol{z}_{\mathrm{stu}}/T_{\mathrm{stu}})\Big), \tag{2}$$

where $\sigma$ denotes the softmax, $T_{\mathrm{tcr}}, T_{\mathrm{stu}} \geq 1$ is the distillation temperature, and $\zeta \in [0,1]$ balances hard versus soft supervision.

# 3 FAILURE MODES IN MODEL MERGING

## 3.1 TASK VECTOR NORM DISPARITY

We begin by demonstrating that differences in task vector norms can severely impair model merging. In practical fine-tuning, practitioners select diverse hyperparameters, including learning rate, number of training steps, weight decay, and optimizer, each of which influences the distance between the final weights and their initialization, i.e. the norm of the task vector.

To quantify this effect, we fine-tuned CLIP models with Vision Transformers (ViTs) backbones, specifically ViT-B-32, on eight vision tasks as introduced in Section 5.1 with two learning rates, $10^{-5}$ (gray) and $10^{-4}$ (blue), and plotted the resulting task vector norms in Figure 1b. Across all tasks, we observe a 5-7× gap between the two settings. Crucially, the difference is not confined to any particular layer: parameter scales diverge consistently throughout the network, as demonstrated in Section E.1.

Figure 1a reports the corresponding merge performance. The gray bars denote the baseline where all eight tasks are fine-tuned with $10^{-5}$, while the blue bars show the average over eight experiments in each of which one task is replaced with a higher learning rate of $10^{-4}$ and the other seven remain unchanged. We measure performance using normalized accuracy. Injecting a single high-norm task vector degrades every merging method, with losses of up to 14%. These results confirm that norm discrepancies pose a fundamental obstacle to robust task vector merging.

The detrimental effect of norm disparity on model merging can be explained with a straightforward theoretical analysis formalized as Proposition 1.

**Proposition 1.** *Let $\boldsymbol{\tau}_1, \boldsymbol{\tau}_2 \in \mathbb{R}^d$ with $\|\boldsymbol{\tau}_2\| > 0$, and define $\delta := \|\boldsymbol{\tau}_1\|/\|\boldsymbol{\tau}_2\|$. Assume $\boldsymbol{\tau}_1 \perp \boldsymbol{\tau}_2$. For $\boldsymbol{\tau}_{\mathrm{merge}} = \boldsymbol{\tau}_1 + \boldsymbol{\tau}_2$,*

$$\cos(\boldsymbol{\tau}_{\mathrm{merge}}, \boldsymbol{\tau}_2) = \frac{1}{\sqrt{1+\delta^2}} \geq 1 - \tfrac{1}{2}\delta^2, \qquad \cos(\boldsymbol{\tau}_{\mathrm{merge}}, \boldsymbol{\tau}_1) = \frac{\delta}{\sqrt{1+\delta^2}} \leq \delta.$$

*Hence, when $\delta \ll 1$, the merge is nearly perfectly aligned with $\boldsymbol{\tau}_2$ while its alignment with $\boldsymbol{\tau}_1$ is at most $O(\delta)$.*

Empirically, task vectors are observed to be approximately orthogonal (Ilharco et al., 2023); assuming orthogonality, we obtain Proposition 1. The proof is given in Appendix B. This result shows that the merged solution almost entirely inherits the directional characteristics of the high-norm task, while the contribution of the low-norm task vanishes up to $O(\delta)$. Under the Neural Tangent Kernel approximation (Jacot et al., 2018) $\Delta f(x) = f(x; \boldsymbol{\theta}_0 + \boldsymbol{\tau}_{\mathrm{merge}}) - f(x; \boldsymbol{\theta}_0) \approx \boldsymbol{\tau}_{\mathrm{merge}}^{\top} \nabla_{\boldsymbol{\theta}} f(x; \boldsymbol{\theta}_0)$, the functional shift from pre-trained model is determined exclusively by the task vector's direction. Thus, the geometric dominance of the high-norm vector implies that the merged model functionally mimics the high-norm task while failing to preserve the low-norm task's knowledge, leading to a severe performance drop. Consequently, such norm disparity can cause a severe drop in performance on the low-norm task and thereby degrade the overall effectiveness of the merged model.

## 3.2 Low-Confidence Source Models

We now show that low confidence constitutes a second, equally damaging failure mode. Paradoxically, models that are well calibrated can be fragile from the perspective of model merging; conversely, we argue that the more overconfident a source model is, the more robust it becomes to merging.

A model's decisiveness can be quantified by the entropy of its predictive distribution. Using the same experimental configuration as in Section 3.1, we replaced the learning-rate manipulation with a single change: turning label smoothing on or off. Figure 1c plots the resulting prediction entropies: the gray bars correspond to training without label smoothing, while the yellow bars use $\alpha = 0.1$. The vertical axis is logarithmic; with label smoothing the entropy increases by up to three orders of magnitude.

Figure 1a (yellow bars) shows how this reduced confidence affects merging. In all algorithms, the normalized accuracy decreases markedly by up to 24% compared to the baseline without smoothing. This degradation exceeds that caused by norm discrepancies in the previous section, underscoring how harmful low-confidence source models can be. In short, routine training choices that alter confidence (e.g. label smoothing, Mixup, focal loss) can induce large swings in post-merge performance. These phenomena can also be supported from a theoretical perspective. (Appendix C)

## 4 Knowledge Distillation for Task Vector Conditioning

Here, we propose **Dis**tillation for **Ta**sk vector **C**onditioning (**DisTaC**) a KD–based preconditioning method that eliminates the harmful effects of individual task vectors during model merging, as identified in Section 3.

### 4.1 Task Vector Norm Conditioning

First, to correct task vector norm disparity, DisTaC harmonizes the norms while preserving single-task accuracy. A naive countermeasure is to adjust the norm by scaling the task vector, i.e. replacing $\tau_t$ with $\kappa_t \tau_t$ using a scalar scaling factor $\kappa_t$. Unfortunately, this constant rescaling offers no guarantee of performance retention and can severely degrade accuracy relative to the pre-merge model.

We therefore propose to recover the lost performance through KD: starting from $\theta_{\mathrm{pre}} + \kappa_t \tau_t$, we treat the pre-merge model as the teacher and distill its predictions into the rescaled student using only unlabeled data from the same task as the one underlying $\tau_t$. Since DisTaC relies solely on unlabeled data, it uses soft-target distillation only, i.e., we fix $\zeta = 1$ in Eq. 2, omitting the cross-entropy loss entirely.

---

**Algorithm 1** DisTaC

**Require:** Pre-trained parameters $\theta_{\mathrm{pre}}$, task vector $\tau_t$, scaling factor $\kappa_t$, temperature pair $(T_{\mathrm{tcr}}, T_{\mathrm{stu}})$, regularization weight $\beta$, unlabeled dataset $\tilde{\mathcal{D}}_t^u$ drawn from the distribution of task $t$, learning rate $\eta$, number of steps $K$

**Ensure:** Fine-tuned student parameters $\theta$
1: $\theta_0 \leftarrow \theta_{\mathrm{pre}} + \kappa_t \tau_t$ ▷ Anchor point
2: $\theta \leftarrow \theta_0$ ▷ Student initialization
3: **for** $k = 1, 2, \ldots, K$ **do**
4:      Sample mini-batch $\mathcal{B}_t \subset \tilde{\mathcal{D}}_t^u$
5:      $L \leftarrow 0$
6:      **for all** $x_t \in \mathcal{B}_t$ **do**
7:          $z_{\mathrm{tcr}} \leftarrow f(x_t; \theta_{\mathrm{pre}} + \tau_t)$
8:          $z_{\mathrm{stu}} \leftarrow f(x_t; \theta)$
9:          $s_{\mathrm{tcr}} \leftarrow \sigma(z_{\mathrm{tcr}}/T_{\mathrm{tcr}})$
10:         $s_{\mathrm{stu}} \leftarrow \sigma(z_{\mathrm{stu}}/T_{\mathrm{stu}})$
11:         $L \leftarrow L + T_{\mathrm{tcr}} T_{\mathrm{stu}} \mathrm{KL}(s_{\mathrm{tcr}} \,\|\, s_{\mathrm{stu}})$
12:      **end for**
13:      $L \leftarrow \frac{L}{|\mathcal{B}_t|} + \beta \|\theta - \theta_0\|_2^2$
14:      $\theta \leftarrow \theta - \eta \nabla_\theta L$ ▷ Gradient step
15: **end for**

---

Although one might instead fine-tune $\theta_{\mathrm{pre}} + \kappa_t \tau_t$ with labeled examples, obtaining a sufficiently large supervised corpus at merge time is typically impractical. By contrast, access to unlabeled data is commonly assumed during model merging (Yang et al., 2024b; Yan et al., 2025; Yoshida et al., 2025), and KD imposes only mild additional requirements.

To prevent the task vector norm from drifting far from $\theta_{\mathrm{pre}} + \kappa_t \tau_t$ during KD, we include an $\ell_2$ regularizer on their difference, as shown in Algorithm 1.

Table 1: **Comparison of post-merge accuracy across fine-tuning configurations and the effect of DisTaC.** Absolute accuracy is displayed in a large font size, whereas normalized accuracy appears in parentheses in a smaller font. "Individual" denotes the average performance of the source models on their respective tasks, and "MTL" represents the performance of conventional MTL. When the task vector norms diverge (Norm Mismatch) or the source models exhibit low confidence (Low Confidence), performance consistently degrades relative to the standard benchmark setting (Original). Under these conditions, DisTaC effectively pre-conditions the source models, achieving performance comparable to Original even in both stringent settings.

| Method | | Original | | Norm Mismatch | | Low Confidence | |
|---|---|---|---|---|---|---|---|
| | | ViT-B-32 | ViT-L-14 | ViT-B-32 | ViT-L-14 | ViT-B-32 | ViT-L-14 |
| Pre-trained | | 47.3 | 65.1 | 47.3 | 65.1 | 47.3 | 65.1 |
| Individual | | 89.9 | 93.7 | 89.3 | 93.3 | 89.8 | 94.0 |
| MTL | | 87.8 | 92.6 | - | - | - | - |
| Task arithmetic | | 70.4 (78.0) | 84.0 (89.3) | 63.6 (71.8) | 78.6 (84.2) | 51.0 (58.3) | 66.9 (71.5) |
| Task arithmetic | + **DisTaC** | - | - | **70.0** (78.2) | **83.9** (89.6) | **63.6** (72.2) | **77.6** (83.3) |
| TIES | | 74.0 (82.0) | 85.0 (91.9) | 59.1 (66.4) | 74.0 (79.5) | 54.5 (62.0) | 68.3 (73.0) |
| TIES | + **DisTaC** | - | - | **73.1** (81.0) | **84.4** (90.2) | **68.7** (77.9) | **79.4** (85.4) |
| Consensus TA | | 74.8 (82.8) | 85.3 (90.7) | 68.8 (77.0) | 82.0 (87.6) | 54.6 (62.0) | 68.6 (73.2) |
| Consensus TA | + **DisTaC** | - | - | **73.7** (82.2) | **84.9** (90.7) | **67.7** (76.5) | **80.0** (85.8) |
| EMR-Merging | | 88.5 (98.4) | 93.0 (99.6) | 80.0 (88.7) | 87.6 (93.6) | 39.2 (45.1) | 27.4 (30.1) |
| EMR-Merging | + **DisTaC** | - | - | **88.1** (97.3) | **92.7** (99.0) | **70.3** (79.2) | **92.3** (98.1) |
| TSVM | | 83.3 (92.4) | 90.5 (96.3) | 72.2 (80.2) | 84.8 (90.7) | 60.7 (68.4) | 71.6 (76.4) |
| TSVM | + **DisTaC** | - | - | **82.9** (91.8) | **90.3** (96.6) | **81.5** (91.8) | **89.7** (96.2) |
| Iso-CTS | | 81.0 (89.7) | 90.4 (96.4) | 78.1 (86.2) | 90.8 (96.9) | 72.5 (81.1) | 80.8 (86.0) |
| Iso-CTS | + **DisTaC** | - | - | **80.3** (88.9) | 90.1 (96.1) | 69.0 (78.1) | **86.1** (91,5) |
| WUDI-Merging | | 85.5 (93.9) | 91.7 (97.7) | 49.2 (52.6) | 57.9 (60.8) | 38.0 (40.8) | 28.0 (29.2) |
| WUDI-Merging | + **DisTaC** | - | - | **84.4** (93.2) | **91.4** (97.5) | **73.8** (83.3) | **91.6** (97.3) |

## 4.2 SOURCE MODEL CONFIDENCE CONDITIONING

To mitigate low-confidence issues, DisTaC aims to increase each source model's confidence before merging, thereby rendering the model more robust to the merge. Here the student and the teacher are identical at initialization, i.e. $\theta_t = \theta_{\mathrm{pre}} + \tau_t$. We set the student temperature $T_{\mathrm{stu}}$ higher than the teacher temperature $T_{\mathrm{tcr}}$ so that the student, trained on a higher-entropy distribution, is pushed toward a lower-entropy (more confident) output when the temperature is later reset to 1. Consequently, the distilled student becomes more confident than its teacher.

One may worry that the over-confidence harms model reliability in practice. However, standard post-hoc calibration methods (e.g. temperature scaling) can mitigate over-confidence, whereas merging with an underconfident model leads to large performance drops that make the merged model impractical. A detailed discussion appears in Section 6.2.

**Unified algorithm.** The two conditioning strategies above are unified by Algorithm 1. When both norm disparity and low-confidence coexist, they can be mitigated simultaneously by choosing an appropriate scaling factor $\kappa_t$ and temperature pair $(T_{\mathrm{tcr}}, T_{\mathrm{stu}})$.

## 5 EXPERIMENT

### 5.1 SETUP

We conducted experiments in a multi-task setting following Ilharco et al. (2023). Specifically, we adopted eight vision tasks: Cars (Krause et al., 2013), DTD (Cimpoi et al., 2014), EuroSAT (Helber et al., 2019), GTSRB (Stallkamp et al., 2011), MNIST (LeCun, 1998), RESISC45 (Cheng et al., 2017), SUN397 (Xiao et al., 2016), and SVHN (Netzer et al., 2011). Our models applied ViT-B-32 and ViT-L-14 to CLIP. We evaluated post-merge performance using absolute accuracy and normalized accuracy under the two aforementioned failure modes: the case with diverged task vector norms (Norm Mismatch) and the case with low-confidence source models (Low Confidence).

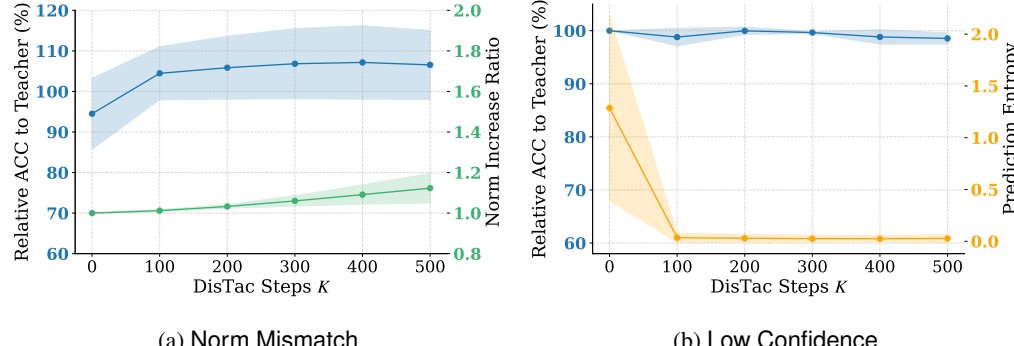

(a) Norm Mismatch                    (b) Low Confidence

Figure 2: **Evolution of DisTaC over steps.** Results are averaged over the eight vision tasks with ViT-B-32; the error band shows one standard deviation around the mean. **(a) Norm Mismatch:** the blue curve plots normalized test accuracy relative to the teacher, and the green curve shows the percentage change in the task vector norm from the DisTaC initialization. Within roughly 100 steps, accuracy recovers to (or exceeds) the teacher's level while the task vector norm remains virtually unchanged from its $\kappa_t$-adjusted target. **(b) Low Confidence:** the blue curve again reports normalized test accuracy, whereas the orange curve tracks the test prediction entropy. About 100 steps suffice to drive the entropy substantially lower, yet the teacher-level accuracy is fully preserved.

The detailed settings for each scenario followed those described in Section 3. We adopted seven merging methods as baselines: task arithmetic (Ilharco et al., 2023), Ties-Merging (TIES) (Yadav et al., 2023), Consensus Merging (Consensus TA) (Wang et al., 2024), EMR-Merging (Huang et al., 2024), TSVM (Gargiulo et al., 2025), Iso-Merging (Iso-CTS) (Marczak et al., 2025), and WUDI-Merging (Cheng et al., 2025). For DisTaC, knowledge distillation was run for $K = 500$ steps. In the Norm Mismatch regime we assign a task–specific scaling coefficient $\kappa_t$ individually for each of the eight norm–disparity configurations: the task vector with the largest $\ell_2$-norm is rescaled so that, after scaling, its norm equals the mean norm of the remaining seven task vectors. A neutral temperature pair is then used, $(T_{\text{tcr}}, T_{\text{stu}}) = (10, 10)$. In the Low Confidence regime we instead fix $\kappa_t = 1$ and sharpen the student by adopting a more asymmetric temperature pair, $(T_{\text{tcr}}, T_{\text{stu}}) = (1, 10)$. More detailed settings can be found in Appendix D.

## 5.2 RESULTS

### 5.2.1 MERGING PERFORMANCE

Table 1 summarizes the results. Absolute accuracy is displayed in a larger font, whereas normalized accuracy appears in parentheses in a smaller font. As noted in Section 3, all methods exhibit a substantial and consistent performance decline relative to the conventional configuration (Original) under both failure modes, revealing a clear vulnerability (white rows). The rows highlighted in gray show the performance obtained by first applying DisTaC for pre-conditioning and then merging. DisTaC consistently enhances merge performance, yielding gains of up to 35.8% absolute accuracy for ViT-B-32 and 63.6% for ViT-L-14. Moreover, for EMR-Merging, which achieves the highest merge performance, DisTaC raises the accuracy under both failure modes to a level comparable with the Original configuration in most cases, indicating that the intended merge performance is robustly maintained even in challenging scenarios.

### 5.2.2 EFFICIENCY OF DISTAC

Here, we present how the single-task performance on each task, the task vector norm, and the prediction entropy change during the KD process of DisTaC, as well as the computational cost required for sufficiently thorough training.

Figure 2 shows the average over eight vision tasks of the training history when KD by DisTaC is applied to ViT-B-32. The blue curve denotes the test accuracy relative to the teacher's test accuracy, the green curve the task vector norm relative to its value at the initialization point, and the orange curve the test prediction entropy.

First, Figure 2a depicts the training history under the Norm Mismatch setting in Table 1. It achieves performance comparable to, or even surpassing, the teacher model's test performance within 500 steps, while the $\ell_2$ regularizer of DisTaC keeps the task vector norm to roughly $1.1\times$ that of the initialization point, $\theta_{\text{pre}} + \kappa_t \tau$, at the end of the 500 steps.

Of particular interest is that DisTaC occasionally surpasses the teacher model's test performance. We identify two factors underlying this phenomenon. The first is the scale given by $\kappa_t$. In particular, we observed that reducing $\kappa_t$ can sometimes improve generalization performance. That is, the DisTaC initialization point already outperforms the teacher model, and we observed this in every instance in which the teacher model was exceeded. This phenomenon of the student outperforming the teacher is confirmed in (Furlanello et al., 2018), where it has been shown that a student can surpass the teacher by repeating KD between identical architectures. Furthermore, in this case, since KD is performed while keeping the student's norm smaller than the teacher's, it is plausible that a regularization effect similar to weight decay is being exhibited.

Next, Figure 2b presents the training history under the Low Confidence setting in Table 1. Within 500 steps, particularly during the first 100 steps, it achieves a substantial reduction in prediction entropy while maintaining test accuracy at a level nearly equivalent to that of the teacher model.

# 6 DISCUSSION

## 6.1 STRETCHING VS. SHRINKING TASK VECTORS

When task vectors differ significantly in norm, a natural question arises: Should shorter vectors be stretched to match longer ones, or should longer vectors be shrunk to match the shorter ones? Our findings support the latter; we advocate shrinking the longer vectors.

There are several reasons for this. First, it is conceivable that model performance is more robust to scaling down a task vector than scaling it up. Figure 3 shows how test accuracy varies across vision tasks when applying different scaling factors $\kappa_t$ to the task vector, i.e., evaluating $\theta_{\text{pre}} + \kappa_t \tau$ for $\kappa_t \in [0.0, 3.0]$. Shrinking the task vector ($\kappa_t < 1.0$) retains performance comparable to or even better than the original fine-tuned model across a broad range. In contrast, stretching beyond $\kappa_t = 1.0$ degrades accuracy, and by $\kappa_t = 3.0$, the model underperforms even the zero-shot baseline across all tasks. A similar trend was also observed for ViT-L-14 (see Section E.3).

As shown earlier in Figure 1b, real-world fine-tuning pipelines often result in over $5\times$ variation in task vector norm due to differing learn-

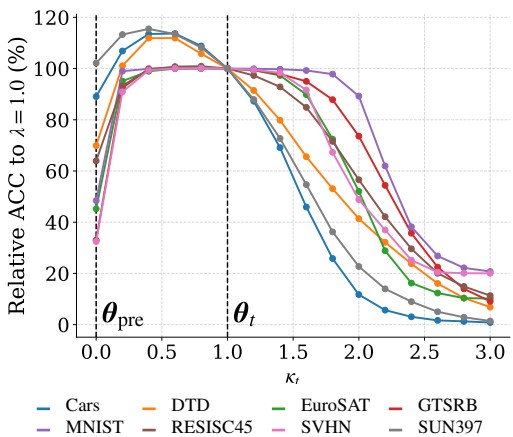

Figure 3: **Effect of scaling task vectors on test accuracy.** For each of the eight vision tasks (ViT-B-32), we evaluate the model $\theta_{\text{pre}} + \kappa_t \tau$ as the scaling factor $\kappa_t$ varies from 0.0 to 3.0. Model performance is more robust to shrinking the task vector than to stretching it, suggesting that when harmonizing task vector norms, longer vectors should be shrunk to match shorter ones.

ing rates or training durations. In such cases, stretching small-norm vectors to match larger ones risks disrupting the pretrained model's useful representations and is therefore undesirable.

Furthermore, Ilharco et al. (2023) observed that merging task vectors with smaller norms tends to yield better performance. A likely explanation is that smaller displacements remain within the local linear regime around $\theta_{\text{pre}}$, where first-order approximations hold more accurately. This also aligns with the NTK perspective discussed in Ortiz-Jimenez et al. (2023); Yoshida et al. (2025), under which merging remains valid and weight disentanglement is preserved near the pretrained initialization. Notably, Theorem 3.1 in Wei et al. (2025a) demonstrates that the performance gap between the merged model and the fine-tuned model is proportional to the product of the learning

rate and the number of fine-tuning steps. This theoretical insight aligns with our claim that shrinking task vectors is preferable.

Taken together, these observations strongly suggest that when normalizing task vectors for merging, it is preferable to shrink the longer ones rather than stretch the shorter ones.

## 6.2 Confidence Reliability in Model Merging

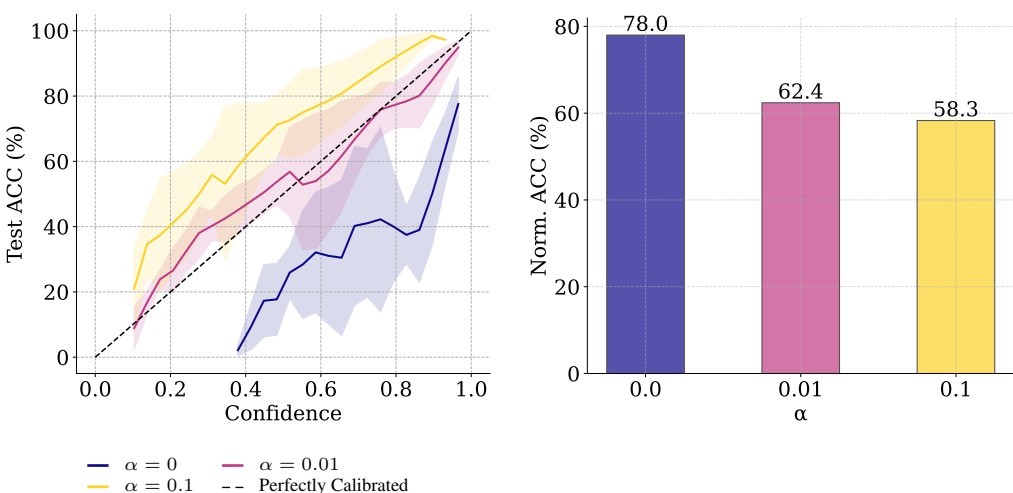

(a) Reliability diagram over different label smoothing strengths

(b) Normalized accuracy over different label smoothing strengths

Figure 4: **Impact of label smoothing on confidence calibration and merge performance.** (a) Average reliability diagram for ViT-B-32 across eight vision tasks under different label-smoothing strengths $\alpha$. Without label smoothing ($\alpha = 0$, dark purple) the model is strongly overconfident; as $\alpha$ increases to 0.01 the model becomes well-calibrated, and at $\alpha = 0.1$ it turns underconfident. (b) Test normalized accuracy obtained when the corresponding source models are merged. Merge performance decreases monotonically with larger $\alpha$, revealing a clear trade-off: lower confidence comes at the cost of lower accuracy after merging.

As noted in Section 3.2, successful model merging often conflicts with maintaining reliable confidence estimates in both the source and merged models. Figure 4 illustrates this trade-off by sweeping the label-smoothing strength $\alpha$ used during fine-tuning of the source models.

First, the calibration curves in Figure 4a show that a model trained without label smoothing (dark-purple line) is strongly overconfident, which is consistent with the well-known tendency of modern deep networks (Guo et al., 2017). As $\alpha$ increases from 0.01 to 0.1 (red → yellow), the models become well-calibrated and eventually underconfident, matching the observations of Müller et al. (2019). Figure 4b then reports the normalized accuracy obtained when these source models are merged. Accuracy decreases monotonically with larger $\alpha$, revealing an inverse correlation between label-smoothing strength and merge performance.

In short, current merging methods perform best when the source models are deliberately overconfident. To retain reliable confidence after merging, we therefore advocate applying post-hoc calibration, such as temperature scaling (Guo et al., 2017), to the merged model rather than trying to calibrate the sources beforehand.

## 7 Limitation

While our main experiments are primarily limited to vision tasks using CLIP, we demonstrate in Appendix 6 the significance of each failure mode and the effectiveness of DiSTaC in NLP tasks. However, since our evaluations in both domains are exclusively restricted to classification tasks, extending our framework to generation tasks and other modalities remains a highly critical direction

for future exploration. Additionally, rather than exploring all possible causes of task interference, we specifically focus on the two main failure modes: norm disparity and low source-model confidence. Furthermore, DisTaC assumes access to unlabeled data for distillation, which can at times be challenging due to potential security constraints. Nevertheless, we emphasize that DisTaC achieves over 96% of ideal performance even when using extremely small datasets or data with severe distribution shifts, demonstrating strong robustness in such settings (see Appendix E.5). Furthermore, other approaches, such as Yang et al. (2024b); Yan et al. (2025), also rely on the availability of unlabeled data. Despite these limitations, we believe that our experiments directly support our main claims on failure modes and are sufficient to demonstrate the effectiveness of our approach.

## 8 CONCLUSION

We presented DisTaC, a lightweight and practical pre-conditioning method for task vectors that improves the robustness of model merging in multi-task learning. Our analysis identified two major failure modes of norm disparity and low source-model confidence that frequently occur in real-world merging scenarios. DisTaC addresses both issues simultaneously via KD on unlabeled data, requiring only minimal computational cost and no access to task labels. Through extensive experiments, we demonstrated that DisTaC not only recovers performance degraded by task vector scaling, but also enhances confidence in the source models without sacrificing generalization. Furthermore, we showed that DisTaC enables state-of-the-art merging methods to succeed in challenging cases where they would otherwise fail. Our findings highlight the importance of task vector conditioning, and we believe that DisTaC provides a simple yet powerful tool to make model merging more reliable and broadly applicable.

## ACKNOWLEDGEMENT

Our deepest gratitude goes out to the anonymous reviewers whose invaluable insights substantially enhanced the quality of this manuscript. This work was supported by RBC Borealis through the RBC Borealis AI Global Fellowship Award, which was awarded to Hiroki Naganuma. The computation resource of this project is supported by "TSUBAME Encouragement Program for Young/Female Users" of Center for Information Infrastructure at Institute of Science Tokyo and by "Joint Usage/Research Center for Interdisciplinary Large-scale Information Infrastructures" in Japan.

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

# A RELATED WORK

## A.1 MODEL MERGING AND TASK ARITHMETIC

Research on integrating multiple neural network models by performing operations on their parameters has been widely conducted, starting with Utans (1996). These techniques enable a model to learn diverse tasks with less time and computational resources, and have become increasingly important in recent years as the number of model parameters has grown dramatically. For instance, in early approaches to model merging, models with the same architecture were fine-tuned and then merged by averaging their parameters (Wortsman et al., 2022a; Choshen et al., 2022). More sophisticated methods have since been proposed, such as Fisher Merging (Matena & Raffel, 2022), which is based on maximizing the posterior probability of the model, and RegMean (Jin et al., 2023), which minimizes the distance between output activations before and after merging. In contrast, task arithmetic (Ilharco et al., 2023) focuses on the task vector, defined as the difference in parameters between a fine-tuned model and a pre-trained model, and performs addition and subtraction of task vectors in parameter space. This approach offers the advantage of allowing flexible, localized modifications to the model and has found applications across diverse tasks (Tang et al., 2024; Su et al., 2024; Yoshikawa et al., 2025; Naganuma et al., 2025).

Recent research on task arithmetic has theoretically analyzed the simple addition of task vectors and proposed multiple methods to address its shortcomings. Approaches aimed at improving the properties of task vectors focus on the linearity in fine-tuning (Ortiz-Jimenez et al., 2023; Yoshida et al., 2025). These methods, based on the Neural Tangent Kernel (NTK) (Jacot et al., 2018), treat the model's output as linear during fine-tuning in order to reflect vector operations in parameter space onto the model's inputs and outputs. Meanwhile, several studies have been conducted from the perspective of mitigating interference between task vectors. TIES-Merging (Yadav et al., 2023) emphasizes the removal of redundant elements and the consideration of sign in each vector dimension. AdaMerging (Yang et al., 2024b), on the other hand, automatically adjusts merging coefficients per task and per layer to reduce task interference and enhance robustness through test-time adaptation. Wang et al. (2024) introduced a framework for pinpointing the parameters that carry information shared across tasks and, on that basis, proposed Consensus Merging, which builds task-wise masks that align more closely with inter-task consensus than the masks used in TIES-Merging. While traditional multi-objective optimization can be computationally prohibitive, Li et al. (2025) amortized this cost by leveraging quadratic approximations to identify diverse Pareto-optimal merging solutions. More recently, Wei et al. (2025b) reformulated model merging as minimizing the loss gap between the merged model and each task-specific model, introducing DOGE with subspace projection and task-aware scaling. TSVM (Gargiulo et al., 2025) interprets task interference as non-orthogonality among the layer-wise singular vectors of the task vectors; by whitening those singular directions, TSVM further improves merge quality.

Despite these advances, Ilharco et al. (2023) and nearly all follow-up studies on multi-task model merging benchmark their methods under highly idealized settings, leaving real-world failure modes largely unexplored. In this work, we show that (i) discrepancies in task vector norms and (ii) low source-model confidence are key sources of interference. We introduce DisTaC as a simple preconditioning step that mitigates both problems before merging.

## A.2 KNOWLEDGE DISTILLATION

DisTaC addresses the limitations of existing task arithmetic methods by incorporating knowledge distillation. Knowledge distillation is a technique proposed for transferring knowledge from a teacher model to a smaller student model (Hinton et al., 2015). Although initially intended for model compression (Hinton et al., 2015; Kim et al., 2018; Sanh et al., 2020), it has also been applied in contexts such as self-distillation, where repeated distillation between models of the same architecture leads to performance improvement (Furlanello et al., 2018; Zhang et al., 2019; Zhang & Sabuncu, 2020). Among these applications, several studies have explored generating models that can handle multiple tasks by distilling knowledge from single or multiple teacher models (Luo et al., 2019; Hao et al., 2023; Xu et al., 2023). These approaches achieve distillation by mapping the parameters of multiple teacher models into a shared space for the student model. Conversely, it is also possible to distill models with different architectures individually to obtain task vectors, which can

then be merged using task arithmetic (Merugu et al., 2025). DisTaC adopts the latter approach and resolves the issue of variability in the norms of task vectors by obtaining them through distillation.

Applying distillation to task arithmetic requires addressing the impact of soft targets. Numerous studies have analyzed the effects of label smoothing in the context of knowledge distillation (Müller et al., 2019; Shen et al., 2021; Chandrasegaran et al., 2022; Zheng & YANG, 2024). In this study, we demonstrate that fine-tuning with soft targets significantly affects the models obtained through model merging, and propose a method to mitigate this effect by increasing the confidence of the student model.

## B   PROOF FOR PROPOSITION 1

Let $\delta = \|\boldsymbol{\tau}_1\|/\|\boldsymbol{\tau}_2\|$ and assume $\boldsymbol{\tau}_1 \perp \boldsymbol{\tau}_2$. Then

$$\|\boldsymbol{\tau}_{\mathrm{merge}}\|^2 = \|\boldsymbol{\tau}_1 + \boldsymbol{\tau}_2\|^2 = \|\boldsymbol{\tau}_1\|^2 + \|\boldsymbol{\tau}_2\|^2 = (1+\delta^2)\|\boldsymbol{\tau}_2\|^2.$$

For the cosine similarity with $\boldsymbol{\tau}_2$, we compute

$$\cos(\boldsymbol{\tau}_{\mathrm{merge}}, \boldsymbol{\tau}_2) = \frac{\boldsymbol{\tau}_{\mathrm{merge}} \cdot \boldsymbol{\tau}_2}{\|\boldsymbol{\tau}_{\mathrm{merge}}\|\|\boldsymbol{\tau}_2\|} = \frac{\|\boldsymbol{\tau}_2\|^2}{\sqrt{(1+\delta^2)}\,\|\boldsymbol{\tau}_2\|^2} = \frac{1}{\sqrt{1+\delta^2}}.$$

Using the inequality $(1+\delta^2)^{-1/2} \geq 1 - \frac{1}{2}\delta^2$ for $\delta \geq 0$, we obtain the lower bound.

Similarly, for the cosine similarity with $\boldsymbol{\tau}_1$,

$$\cos(\boldsymbol{\tau}_{\mathrm{merge}}, \boldsymbol{\tau}_1) = \frac{\boldsymbol{\tau}_{\mathrm{merge}} \cdot \boldsymbol{\tau}_1}{\|\boldsymbol{\tau}_{\mathrm{merge}}\|\|\boldsymbol{\tau}_1\|} = \frac{\|\boldsymbol{\tau}_1\|^2}{\sqrt{(1+\delta^2)}\,\|\boldsymbol{\tau}_1\|\|\boldsymbol{\tau}_2\|} = \frac{\delta}{\sqrt{1+\delta^2}}.$$

Since $\delta/\sqrt{1+\delta^2} \leq \delta$, the claim follows.

Hence, when $\delta \ll 1$, the merged vector is nearly aligned with $\boldsymbol{\tau}_2$ while its alignment with $\boldsymbol{\tau}_1$ is suppressed by a factor of $O(\delta)$. □

## C   THEORETICAL INSIGHTS INTO TASK VECTOR MERGING FOR MODELS OPTIMIZED WITH DISTINCT OBJECTIVES

This appendix provides a step-by-step derivation of the theoretical results concerning the effect of calibration penalties on the arithmetic merging of task vectors. We demonstrate how calibration can introduce a first-order degradation in cross-entropy (CE) performance upon merging, an effect not observed when merging standard CE-trained task vectors.

### C.1   NOTATION AND ASSUMPTIONS

We use the following notation. For task $i$, the standard cross-entropy (CE) objective is

$$J_i^{\mathrm{CE}}(\boldsymbol{\theta}) := -\,\mathbb{E}_{(x,y)\sim\mathcal{D}_i}[\log p_{\boldsymbol{\theta}}(\boldsymbol{y} \mid \boldsymbol{x})]\,.$$

We also consider a calibrated objective that augments CE with a generic penalty $\mathcal{C}_i(\boldsymbol{\theta})$ [1] weighted by $\lambda_i > 0$:

$$J_i^{\mathrm{CAL}}(\boldsymbol{\theta}) := J_i^{\mathrm{CE}}(\boldsymbol{\theta}) + \lambda_i\,\mathcal{C}_i(\boldsymbol{\theta}).$$

For either objective $\star \in \{\mathrm{CE}, \mathrm{CAL}\}$, the task-specific optimum is denoted

$$\boldsymbol{\theta}_i^\star := \arg\min_{\boldsymbol{\theta}} J_i^\star(\boldsymbol{\theta}).$$

Throughout, we assume the objectives $J_i^{\mathrm{CE}}$ and $J_i^{\mathrm{CAL}}$ are $C^2$ in a neighborhood of a fixed base parameter $\boldsymbol{\theta}_0$. Let $H_i := \nabla^2 J_i^{\mathrm{CE}}(\boldsymbol{\theta}_0)$ denote the CE Hessian at $\boldsymbol{\theta}_0$ and assume $H_i$ is positive-definite, ensuring that $\boldsymbol{\theta}_0$ lies in a locally convex region of the CE landscape. For notational convenience we write the gradients at $\boldsymbol{\theta}_0$ as

$$\mathbf{g}_i := \nabla J_i^{\mathrm{CE}}(\boldsymbol{\theta}_0), \qquad \mathbf{b}_i := \nabla \mathcal{C}_i(\boldsymbol{\theta}_0).$$

---

[1]For example, a detailed description of evaluating focal loss can be found in Kimura & Naganuma (2025).

## C.2 THEORETICAL PRELIMINARIES FOR THE MAIN RESULT

### C.2.1 DERIVATION OF THE STANDARD TASK VECTOR

The optimal parameter vector $\boldsymbol{\theta}_i^{\mathrm{CE}}$ for the standard cross-entropy loss satisfies the first-order optimality condition, which states that the gradient at this point is zero.

$$\nabla J_i^{\mathrm{CE}}(\boldsymbol{\theta}_i^{\mathrm{CE}}) = 0. \tag{3}$$

Using the definition of the task vector, we can write $\boldsymbol{\theta}_i^{\mathrm{CE}} = \boldsymbol{\theta}_0 + \boldsymbol{\tau}_i^{\mathrm{CE}}$. Substituting this into the optimality condition yields:

$$\nabla J_i^{\mathrm{CE}}(\boldsymbol{\theta}_0 + \boldsymbol{\tau}_i^{\mathrm{CE}}) = 0. \tag{4}$$

We now perform a first-order Taylor series expansion of the gradient function $\nabla J_i^{\mathrm{CE}}(\cdot)$ around the point $\boldsymbol{\theta}_0$.

$$\nabla J_i^{\mathrm{CE}}(\boldsymbol{\theta}_0 + \boldsymbol{\tau}_i^{\mathrm{CE}}) = \nabla J_i^{\mathrm{CE}}(\boldsymbol{\theta}_0) + \nabla^2 J_i^{\mathrm{CE}}(\boldsymbol{\theta}_0)\boldsymbol{\tau}_i^{\mathrm{CE}} + \mathcal{O}(\|\boldsymbol{\tau}_i^{\mathrm{CE}}\|^2). \tag{5}$$

Using our established notation for the gradient ($\mathbf{g}_i$) and the Hessian ($H_i$) at $\boldsymbol{\theta}_0$, this becomes:

$$\mathbf{g}_i + H_i\boldsymbol{\tau}_i^{\mathrm{CE}} + \mathcal{O}(\|\boldsymbol{\tau}_i^{\mathrm{CE}}\|^2) = 0. \tag{6}$$

For fine-tuning scenarios where the task-specific solution $\boldsymbol{\theta}_i^{\mathrm{CE}}$ is close to the pre-trained model $\boldsymbol{\theta}_0$, the norm of the task vector $\|\boldsymbol{\tau}_i^{\mathrm{CE}}\|$ is small. We can therefore neglect the higher-order terms.

$$\mathbf{g}_i + H_i\boldsymbol{\tau}_i^{\mathrm{CE}} \approx 0. \tag{7}$$

Since $H_i$ is assumed to be positive-definite, it is invertible. We can solve for the task vector $\boldsymbol{\tau}_i^{\mathrm{CE}}$:

$$H_i\boldsymbol{\tau}_i^{\mathrm{CE}} = -\boldsymbol{\tau}_i, \tag{8}$$

which gives the well-known result from a single Newton-Raphson step:

$$\boldsymbol{\tau}_i^{\mathrm{CE}} = -H_i^{-1}\mathbf{g}_i. \tag{9}$$

### C.2.2 DERIVATION OF THE CALIBRATED TASK VECTOR

We now apply the same procedure to the calibrated objective function $J_i^{\mathrm{CAL}}(\boldsymbol{\theta})$.

**Gradient and hessian at the base point.** First, we compute the gradient and Hessian of $J_i^{\mathrm{CAL}}(\boldsymbol{\theta})$ at the base point $\boldsymbol{\theta}_0$. The gradient is:

$$\nabla J_i^{\mathrm{CAL}}(\boldsymbol{\theta}_0) = \nabla \left( J_i^{\mathrm{CE}}(\boldsymbol{\theta}) + \lambda_i \mathcal{C}_i(\boldsymbol{\theta}) \right) \Big|_{\boldsymbol{\theta}=\boldsymbol{\theta}_0} \tag{10}$$

$$= \nabla J_i^{\mathrm{CE}}(\boldsymbol{\theta}_0) + \lambda_i \nabla \mathcal{C}_i(\boldsymbol{\theta}_0) \tag{11}$$

$$= \mathbf{g}_i + \lambda_i \mathbf{b}_i. \tag{12}$$

Let $A_i := \nabla^2 \mathcal{C}_i(\boldsymbol{\theta}_0)$ be the Hessian of the calibration term. The Hessian of the calibrated objective, which we denote by $\tilde{H}_i$, is:

$$\tilde{H}_i := \nabla^2 J_i^{\mathrm{CAL}}(\boldsymbol{\theta}_0) = \nabla^2 \left( J_i^{\mathrm{CE}}(\boldsymbol{\theta}) + \lambda_i \mathcal{C}_i(\boldsymbol{\theta}) \right) \Big|_{\boldsymbol{\theta}=\boldsymbol{\theta}_0} \tag{13}$$

$$= \nabla^2 J_i^{\mathrm{CE}}(\boldsymbol{\theta}_0) + \lambda_i \nabla^2 \mathcal{C}_i(\boldsymbol{\theta}_0) \tag{14}$$

$$= H_i + \lambda_i A_i. \tag{15}$$

**Neumann series expansion of $\tilde{H}_i^{-1}$.** To solve for the calibrated task vector $\boldsymbol{\tau}_i^{\mathrm{CAL}}$, we need the inverse of the calibrated Hessian, $\tilde{H}_i^{-1}$. For a small penalty weight $\lambda_i$, we can approximate this inverse. We begin by factoring out $H_i$:

$$\tilde{H}_i = H_i + \lambda_i A_i = H_i \left( I + H_i^{-1}(\lambda_i A_i) \right) = H_i \left( I + \lambda_i H_i^{-1} A_i \right). \tag{16}$$

The inverse is then given by $\tilde{H}_i^{-1} = (I + \lambda_i H_i^{-1} A_i)^{-1} H_i^{-1}$. We can expand the term $(I + \lambda_i H_i^{-1} A_i)^{-1}$ using a Neumann series (Horn & Johnson, 2012), $(I + X)^{-1} = \sum_{k=0}^{\infty}(-X)^k$,

which converges if the spectral radius of $X$ is less than 1. Assuming $\lambda_i$ is sufficiently small such that $\|\lambda_i H_i^{-1} A_i\| < 1$, we have:

$$(I + \lambda_i H_i^{-1} A_i)^{-1} = I - \lambda_i H_i^{-1} A_i + (\lambda_i H_i^{-1} A_i)^2 - \ldots \tag{17}$$

$$= I - \lambda_i H_i^{-1} A_i + \mathcal{O}(\lambda_i^2). \tag{18}$$

Substituting this back into the expression for $\tilde{H}_i^{-1}$:

$$\tilde{H}_i^{-1} = (I - \lambda_i H_i^{-1} A_i + \mathcal{O}(\lambda_i^2)) H_i^{-1} \tag{19}$$

$$= H_i^{-1} - \lambda_i H_i^{-1} A_i H_i^{-1} + \mathcal{O}(\lambda_i^2). \tag{20}$$

**Solving for $\boldsymbol{\tau}_i^{\mathrm{CAL}}$.** The calibrated task vector $\boldsymbol{\tau}_i^{\mathrm{CAL}}$ is found by applying the first-order optimality condition to $J_i^{\mathrm{CAL}}$ and linearizing around $\boldsymbol{\theta}_0$:

$$\nabla J_i^{\mathrm{CAL}}(\boldsymbol{\theta}_i^{\mathrm{CAL}}) = \nabla J_i^{\mathrm{CAL}}(\boldsymbol{\theta}_0) + \nabla^2 J_i^{\mathrm{CAL}}(\boldsymbol{\theta}_0) \boldsymbol{\tau}_i^{\mathrm{CAL}} + \mathcal{O}(\|\boldsymbol{\tau}_i^{\mathrm{CAL}}\|^2) = 0. \tag{21}$$

Using the expressions from B1 and ignoring higher-order terms:

$$(\mathbf{g}_i + \lambda_i \mathbf{b}_i) + \tilde{H}_i \boldsymbol{\tau}_i^{\mathrm{CAL}} \approx 0. \tag{22}$$

Solving for $\boldsymbol{\tau}_i^{\mathrm{CAL}}$ gives:

$$\boldsymbol{\tau}_i^{\mathrm{CAL}} \approx -\tilde{H}_i^{-1}(\mathbf{g}_i + \lambda_i \mathbf{b}_i). \tag{23}$$

Now, we substitute the approximation for $\tilde{H}_i^{-1}$ from equation 20:

$$\boldsymbol{\tau}_i^{\mathrm{CAL}} \approx -\left(H_i^{-1} - \lambda_i H_i^{-1} A_i H_i^{-1} + \mathcal{O}(\lambda_i^2)\right)(\mathbf{g}_i + \lambda_i \mathbf{b}_i) \tag{24}$$

$$= -\left(H_i^{-1}\mathbf{g}_i + \lambda_i H_i^{-1}\mathbf{b}_i - \lambda_i H_i^{-1} A_i H_i^{-1}\mathbf{g}_i - \lambda_i^2 H_i^{-1} A_i H_i^{-1}\mathbf{b}_i\right) + \mathcal{O}(\lambda_i^2) \tag{25}$$

$$= -H_i^{-1}\mathbf{g}_i - \lambda_i H_i^{-1}\mathbf{b}_i + \lambda_i H_i^{-1} A_i H_i^{-1}\mathbf{g}_i + \mathcal{O}(\lambda_i^2). \tag{26}$$

We recognize the first term as the standard task vector, $\boldsymbol{\tau}_i^{\mathrm{CE}} = -H_i^{-1}\mathbf{g}_i$. The expression becomes:

$$\boldsymbol{\tau}_i^{\mathrm{CAL}} = \boldsymbol{\tau}_i^{\mathrm{CE}} - \lambda_i H_i^{-1}\mathbf{b}_i + \lambda_i H_i^{-1} A_i H_i^{-1}\mathbf{g}_i + \mathcal{O}(\lambda_i^2). \tag{27}$$

In many practical scenarios, especially after extensive pre-training, the initial gradient norm $\|\mathbf{g}_i\|$ is small. Consequently, the term $\lambda_i H_i^{-1} A_i H_i^{-1}\mathbf{g}_i$, which is of order $\mathcal{O}(\lambda_i\|\mathbf{g}_i\|)$, is often negligible compared to the term $-\lambda_i H_i^{-1}\mathbf{b}_i$, which is $\mathcal{O}(\lambda_i)$. Under this simplifying assumption, we can define the first-order correction due to calibration as:

$$\boldsymbol{\delta}_i := -\lambda_i H_i^{-1}\mathbf{b}_i. \tag{28}$$

This allows us to express the calibrated task vector as a simple perturbation of the standard task vector:

$$\boldsymbol{\tau}_i^{\mathrm{CAL}} = \boldsymbol{\tau}_i^{\mathrm{CE}} + \boldsymbol{\delta}_i + \mathcal{O}(\lambda_i^2, \lambda_i\|\mathbf{g}_i\|). \tag{29}$$

### C.2.3 Task Vector Merging

We consider merging two task vectors using a simple linear combination with positive coefficients $\alpha, \beta > 0$. We define two types of merged parameters:

$$\boldsymbol{\theta}_{\mathrm{merge}}^{\mathrm{CE}} := \boldsymbol{\theta}_0 + \alpha \boldsymbol{\tau}_1^{\mathrm{CE}} + \beta \boldsymbol{\tau}_2^{\mathrm{CE}}, \tag{30}$$

$$\boldsymbol{\theta}_{\mathrm{merge}}^{\mathrm{CAL}} := \boldsymbol{\theta}_0 + \alpha \boldsymbol{\tau}_1^{\mathrm{CAL}} + \beta \boldsymbol{\tau}_2^{\mathrm{CAL}}. \tag{31}$$

**Taylor expansion of the CE loss for merged vectors.** Our goal is to evaluate the CE loss $J_i^{\mathrm{CE}}$ not at its own optimum, but at the merged parameter points. We use a second-order Taylor expansion of $J_i^{\mathrm{CE}}(\boldsymbol{\theta})$ around $\boldsymbol{\theta}_0$:

$$J_i^{\mathrm{CE}}(\boldsymbol{\theta}) - J_i^{\mathrm{CE}}(\boldsymbol{\theta}_0) = \mathbf{g}_i^\top (\boldsymbol{\theta} - \boldsymbol{\theta}_0) + \frac{1}{2}(\boldsymbol{\theta} - \boldsymbol{\theta}_0)^\top H_i (\boldsymbol{\theta} - \boldsymbol{\theta}_0) + \mathcal{O}(\|\boldsymbol{\theta} - \boldsymbol{\theta}_0\|^3). \tag{32}$$

**Merging of CE vectors.** Let $\Delta\boldsymbol{\theta}^{\mathrm{CE}} = \boldsymbol{\theta}^{\mathrm{CE}}_{\mathrm{merge}} - \boldsymbol{\theta}_0 = \alpha\boldsymbol{\tau}^{\mathrm{CE}}_1 + \beta\boldsymbol{\tau}^{\mathrm{CE}}_2$. The change in CE loss for task $i$ is:

$$J^{\mathrm{CE}}_i(\boldsymbol{\theta}^{\mathrm{CE}}_{\mathrm{merge}}) - J^{\mathrm{CE}}_i(\boldsymbol{\theta}_0) = \mathbf{g}^{\top}_i(\alpha\boldsymbol{\tau}^{\mathrm{CE}}_1 + \beta\boldsymbol{\tau}^{\mathrm{CE}}_2) + \mathcal{O}(\|\boldsymbol{\tau}\|^2). \tag{33}$$

Let's analyze the first-order term in the expansion. Using $\mathbf{g}_i = -H_i\boldsymbol{\tau}^{\mathrm{CE}}_i$ from equation 7:

$$\mathbf{g}^{\top}_i(\alpha\boldsymbol{\tau}^{\mathrm{CE}}_1 + \beta\boldsymbol{\tau}^{\mathrm{CE}}_2) = \alpha\mathbf{g}^{\top}_i\boldsymbol{\tau}^{\mathrm{CE}}_1 + \beta\mathbf{g}^{\top}_i\boldsymbol{\tau}^{\mathrm{CE}}_2 \tag{34}$$

$$= \alpha(-H_i\boldsymbol{\tau}^{\mathrm{CE}}_i)^{\top}\boldsymbol{\tau}^{\mathrm{CE}}_1 + \beta(-H_i\boldsymbol{\tau}^{\mathrm{CE}}_i)^{\top}\boldsymbol{\tau}^{\mathrm{CE}}_2 \tag{35}$$

$$= -\alpha(\boldsymbol{\tau}^{\mathrm{CE}}_i)^{\top}H_i\boldsymbol{\tau}^{\mathrm{CE}}_1 - \beta(\boldsymbol{\tau}^{\mathrm{CE}}_i)^{\top}H_i\boldsymbol{\tau}^{\mathrm{CE}}_2. \tag{36}$$

The term for task $i$ itself ($i = 1$ and analyzing $\boldsymbol{\tau}^{\mathrm{CE}}_1$, or $i = 2$ and analyzing $\boldsymbol{\tau}^{\mathrm{CE}}_2$) is $-\alpha(\boldsymbol{\tau}^{\mathrm{CE}}_i)^{\top}H_i\boldsymbol{\tau}^{\mathrm{CE}}_i = -\alpha\|\boldsymbol{\tau}^{\mathrm{CE}}_i\|^2_{H_i}$. Since $H_i$ is positive-definite, this self-term is strictly negative. The cross-term's sign is indefinite. However, the dominant contribution to the loss change is typically negative and of order $\mathcal{O}(\|\boldsymbol{\tau}\|^2)$, indicating that merging CE vectors does not increase the loss at first order.

**Merging of calibrated vectors.** Let $\Delta\boldsymbol{\theta}^{\mathrm{CAL}} = \boldsymbol{\theta}^{\mathrm{CAL}}_{\mathrm{merge}} - \boldsymbol{\theta}_0 = \alpha\boldsymbol{\tau}^{\mathrm{CAL}}_1 + \beta\boldsymbol{\tau}^{\mathrm{CAL}}_2$. The change in loss is:

$$J^{\mathrm{CE}}_i(\boldsymbol{\theta}^{\mathrm{CAL}}_{\mathrm{merge}}) - J^{\mathrm{CE}}_i(\boldsymbol{\theta}_0) = \mathbf{g}^{\top}_i(\alpha\boldsymbol{\tau}^{\mathrm{CAL}}_1 + \beta\boldsymbol{\tau}^{\mathrm{CAL}}_2) + \mathcal{O}(\|\boldsymbol{\tau}\|^2, \lambda^2). \tag{37}$$

We substitute $\boldsymbol{\tau}^{\mathrm{CAL}}_j \approx \boldsymbol{\tau}^{\mathrm{CE}}_j + \boldsymbol{\delta}_j$:

$$\mathbf{g}^{\top}_i(\alpha\boldsymbol{\tau}^{\mathrm{CAL}}_1 + \beta\boldsymbol{\tau}^{\mathrm{CAL}}_2) \approx \mathbf{g}^{\top}_i\left(\alpha(\boldsymbol{\tau}^{\mathrm{CE}}_1 + \boldsymbol{\delta}_1) + \beta(\boldsymbol{\tau}^{\mathrm{CE}}_2 + \boldsymbol{\delta}_2)\right) \tag{38}$$

$$= \underbrace{\mathbf{g}^{\top}_i(\alpha\boldsymbol{\tau}^{\mathrm{CE}}_1 + \beta\boldsymbol{\tau}^{\mathrm{CE}}_2)}_{\text{Original term, } \mathcal{O}(\|\boldsymbol{\tau}\|^2)} + \underbrace{\alpha(\mathbf{g}^{\top}_i\boldsymbol{\delta}_1) + \beta(\mathbf{g}^{\top}_i\boldsymbol{\delta}_2)}_{\text{Additional term, } \mathcal{O}(\lambda\|\boldsymbol{\tau}\|)}. \tag{39}$$

Let's analyze the additional term introduced by calibration. Using the definitions of $\mathbf{g}_i$ and $\boldsymbol{\delta}_j$:

$$\mathbf{g}^{\top}_i\boldsymbol{\delta}_j = (-H_i\boldsymbol{\tau}^{\mathrm{CE}}_i)^{\top}(-\lambda_jH^{-1}_j\mathbf{b}_j) = \lambda_j(\boldsymbol{\tau}^{\mathrm{CE}}_i)^{\top}H_iH^{-1}_j\mathbf{b}_j. \tag{40}$$

This term is first-order in $\lambda_j$ and its sign is not guaranteed to be negative. If this term is positive, it can cause an increase in the CE loss. Since its magnitude is $\mathcal{O}(\lambda\|\boldsymbol{\tau}\|)$, it can dominate the $\mathcal{O}(\|\boldsymbol{\tau}\|^2)$ terms when $\|\boldsymbol{\tau}\|$ is small, leading to a net increase in the CE loss.

## C.3 MAIN RESULT AND PROOF

**Proposition 2.** *Under the assumptions stated, if the vectors $\{\mathbf{g}^{\top}_i\boldsymbol{\delta}_j\}_{j=1,2}$ are not both zero or strictly negative, then there exist merge coefficients $\alpha, \beta > 0$ such that for at least one task $i \in \{1, 2\}$,*

$$J^{\mathrm{CE}}_i(\boldsymbol{\theta}^{\mathrm{CAL}}_{\mathrm{merge}}) > J^{\mathrm{CE}}_i(\boldsymbol{\theta}^{\mathrm{CE}}_{\mathrm{merge}}).$$

*This difference is of first order in the calibration weights $\lambda_1, \lambda_2$.*

*Proof.* We analyze the difference in the CE loss for task $i$ between the two merging strategies. Let $\Delta\boldsymbol{\theta}^{\mathrm{CE}} = \boldsymbol{\theta}^{\mathrm{CE}}_{\mathrm{merge}} - \boldsymbol{\theta}_0$ and $\Delta\boldsymbol{\theta}^{\mathrm{CAL}} = \boldsymbol{\theta}^{\mathrm{CAL}}_{\mathrm{merge}} - \boldsymbol{\theta}_0$.

$$J^{\mathrm{CE}}_i(\boldsymbol{\theta}^{\mathrm{CAL}}_{\mathrm{merge}}) - J^{\mathrm{CE}}_i(\boldsymbol{\theta}^{\mathrm{CE}}_{\mathrm{merge}})$$
$$= \left(J^{\mathrm{CE}}_i(\boldsymbol{\theta}_0) + \mathbf{g}^{\top}_i\Delta\boldsymbol{\theta}^{\mathrm{CAL}} + \mathcal{O}(\|\Delta\boldsymbol{\theta}^{\mathrm{CAL}}\|^2)\right) - \left(J^{\mathrm{CE}}_i(\boldsymbol{\theta}_0) + \mathbf{g}^{\top}_i\Delta\boldsymbol{\theta}^{\mathrm{CE}} + \mathcal{O}(\|\Delta\boldsymbol{\theta}^{\mathrm{CE}}\|^2)\right)$$
$$= \mathbf{g}^{\top}_i(\Delta\boldsymbol{\theta}^{\mathrm{CAL}} - \Delta\boldsymbol{\theta}^{\mathrm{CE}}) + \mathcal{O}(\|\boldsymbol{\tau}\|^2, \lambda^2). \tag{41}$$

The difference between the merged displacement vectors is:

$$\Delta\boldsymbol{\theta}^{\mathrm{CAL}} - \Delta\boldsymbol{\theta}^{\mathrm{CE}} = \left(\alpha\boldsymbol{\tau}^{\mathrm{CAL}}_1 + \beta\boldsymbol{\tau}^{\mathrm{CAL}}_2\right) - \left(\alpha\boldsymbol{\tau}^{\mathrm{CE}}_1 + \beta\boldsymbol{\tau}^{\mathrm{CE}}_2\right)$$
$$= \alpha(\boldsymbol{\tau}^{\mathrm{CAL}}_1 - \boldsymbol{\tau}^{\mathrm{CE}}_1) + \beta(\boldsymbol{\tau}^{\mathrm{CAL}}_2 - \boldsymbol{\tau}^{\mathrm{CE}}_2)$$
$$= \alpha(\boldsymbol{\delta}_1 + \mathcal{O}(\lambda^2_1)) + \beta(\boldsymbol{\delta}_2 + \mathcal{O}(\lambda^2_2))$$
$$= \alpha\boldsymbol{\delta}_1 + \beta\boldsymbol{\delta}_2 + \mathcal{O}(\lambda^2). \tag{42}$$

Substituting this back, the leading term of the loss difference is:

$$J^{\mathrm{CE}}_i(\boldsymbol{\theta}^{\mathrm{CAL}}_{\mathrm{merge}}) - J^{\mathrm{CE}}_i(\boldsymbol{\theta}^{\mathrm{CE}}_{\mathrm{merge}}) \approx \alpha(\mathbf{g}^{\top}_i\boldsymbol{\delta}_1) + \beta(\mathbf{g}^{\top}_i\boldsymbol{\delta}_2). \tag{43}$$

The terms $\mathbf{g}_i^\top \boldsymbol{\delta}_1$ and $\mathbf{g}_i^\top \boldsymbol{\delta}_2$ are scalars of order $\mathcal{O}(\lambda\|\boldsymbol{\tau}\|)$. Their signs depend on the geometry of the loss landscapes. Unless both scalars are non-positive for both tasks $i = 1, 2$, we can choose positive coefficients $\alpha, \beta$ that result in a positive sum for at least one task. For instance, if $\mathbf{g}_i^\top \boldsymbol{\delta}_1 > 0$ for a given $i$, we can select a small enough $\beta > 0$ relative to $\alpha > 0$ such that the total sum $\alpha(\mathbf{g}_i^\top \boldsymbol{\delta}_1) + \beta(\mathbf{g}_i^\top \boldsymbol{\delta}_2)$ is positive.

This positive term is of order $\mathcal{O}(\lambda\|\boldsymbol{\tau}\|)$. It dominates the other terms of order $\mathcal{O}(\|\boldsymbol{\tau}\|^2)$ and $\mathcal{O}(\lambda^2)$ when $\|\boldsymbol{\tau}\|$ and $\lambda$ are sufficiently small, leading to a net increase in the CE loss for calibrated merging compared to standard merging. $\qquad\square$

**Interpretation**  This result provides a theoretical basis for the observation that merging task vectors trained with certain penalties can be detrimental. The calibration penalty introduces a linear perturbation term $\boldsymbol{\delta}_i$ to the task vector. This term is not necessarily aligned with the descent direction of the cross-entropy loss $J_i^{\mathrm{CE}}$. When multiple such vectors are added, these misaligned perturbations can combine constructively to push the merged parameter vector into a region of higher CE loss. This increase is of first order in $\lambda$ and can therefore be significant. In contrast, merging pure CE vectors does not introduce such a first-order degradation term.

## D  EXPERIMENT DETAILS

All experiments were run on NVIDIA A100 GPUs (40 GB memory each). Fine-tuning jobs used four GPUs in parallel, whereas all evaluations were performed on a single GPU.

**Fine-tuning Details.**  Our training protocol closely mirrors the public code of Ilharco et al. (2023). For each task, we fine-tuned CLIP backbones (ViT-B-32 and ViT-L-14) for 2000 updates using AdamW (Loshchilov & Hutter, 2019) with a weight-decay factor of 0.1. We adopted a cosine-annealed learning-rate schedule preceded by 200 warm-up steps and used a mini-batch size of 128; ViT-L-14 training employed gradient accumulation to match this effective batch size. Following the findings of Ilharco et al. (2023), we kept CLIP's text encoder frozen and treated the logits obtained from class-specific prompts (e.g., "a photo of a {classname}") as a fixed classification head, updating only the image encoder during fine-tuning. Regarding the learning rate, we used $10^{-4}$ only when training task vectors with large norms in the **Norm Mismatch** setting, and $10^{-5}$ for all other cases. In the **Low Confidence** setting, the label smoothing strength was set to $\alpha = 0.1$.

**Merging Details.**  For all four merging methods adopted in this study, it is necessary to tune the task vector coefficient $\lambda_t$. Following Ilharco et al. (2023), we imposed a unified constraint on all $\lambda_t$ and searched the range from 0.0 to 1.0 (in increments such as 0.05) based on validation accuracy.

**Distillation Details.**  The distillation procedure generally followed the fine-tuning settings described above, except that the number of steps was set to 500 and the learning rate was fixed at $10^{-5}$ for all cases. The $\ell_2$ regularizer weight $\beta$ was set to 0.5.

### D.1  NORMALIZED ACCURACY

The normalized accuracy for a task $t$ on its dataset $\tilde{\mathcal{D}}_t$ is defined as the ratio of the post-merge model's accuracy to the single-task model's accuracy:

$$\text{normalized accuracy}_t = \frac{\text{accuracy}(\boldsymbol{\theta}_{\mathrm{mtl}}, \tilde{\mathcal{D}}_t)}{\text{accuracy}(\boldsymbol{\theta}_t, \tilde{\mathcal{D}}_t)},$$

where the function $\text{accuracy}(\boldsymbol{\theta}, \mathcal{D})$ denotes the accuracy of the model $f(\cdot; \boldsymbol{\theta})$ on a dataset $\mathcal{D}$.

## E  ADDITIONAL RESULTS

### E.1  NORM COMPARISON ACROSS LAYERS

Figure 5 (weights) and Figure 6 (biases) visualize how the parameter norm of each ViT-B-32 layer changes when the learning rate is raised from $10^{-5}$ (gray bars) to $10^{-4}$ (blue bars). The scale shift

is not confined to a few layers; rather, every block exhibits a consistent multiplicative increase. In other words, tuning with a larger learning rate stretches the entire task vector almost uniformly, across both weight matrices and bias terms. This layer-wise coherence implies that any merge-time correction must adjust the global scale of the model, not merely a subset of layers.

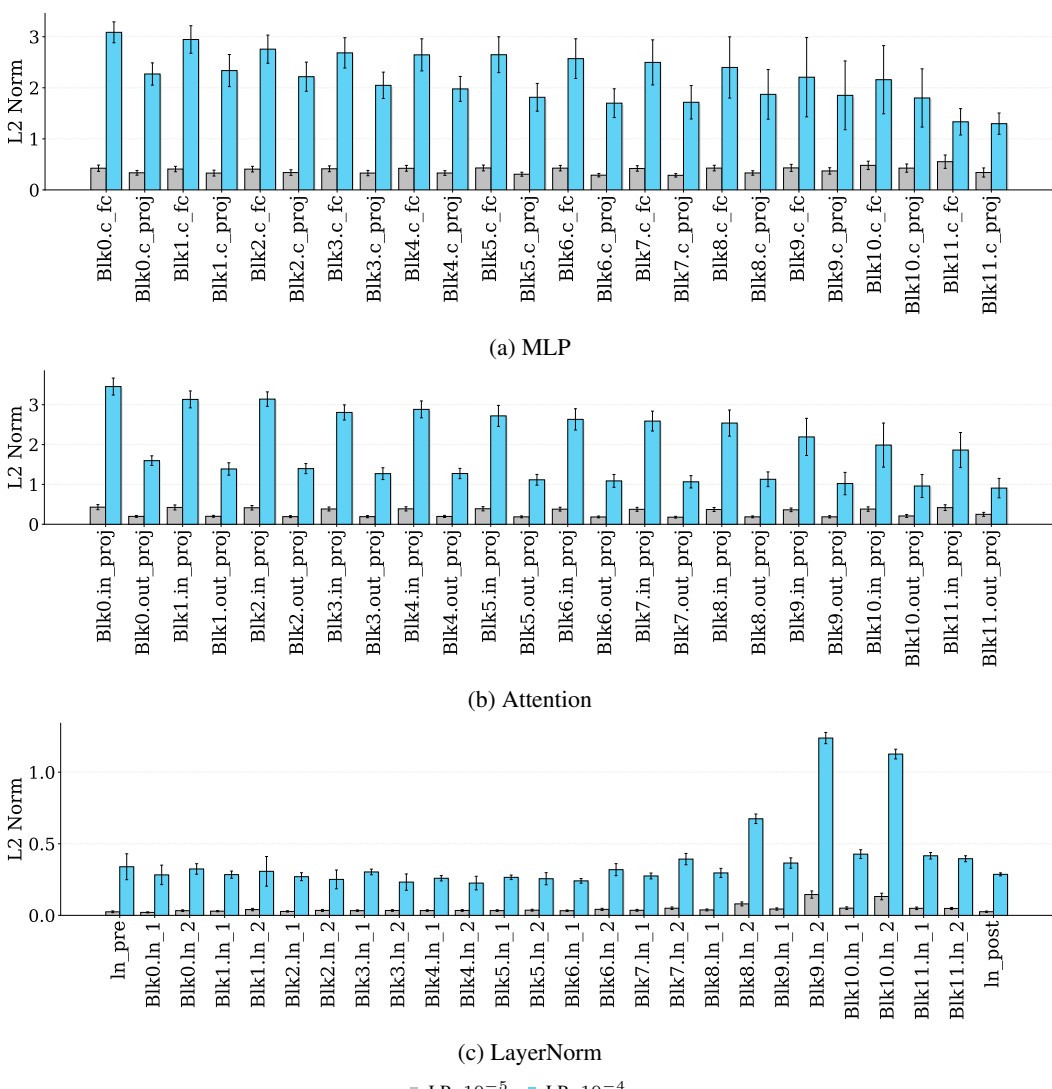

(a) MLP

(b) Attention

(c) LayerNorm

$\blacksquare$ LR=$10^{-5}$ $\blacksquare$ LR=$10^{-4}$

Figure 5: **Layer-wise average task-vector norms for weight parameters in ViT-B-32, averaged over eight vision tasks.** Gray bars correspond to a fine-tuning learning rate of $10^{-5}$, blue bars to $10^{-4}$.

### E.2 OTHER CONFIDENCE CALIBRATION METHOD AND MERGING PERFORMANCE

We assessed two additional confidence–calibration techniques—Mixup and focal loss—alongside label smoothing. For each of the eight vision tasks we fine-tuned ViT-B-32 with Mixup or focal loss and then merged the resulting task vectors. For Mixup, the interpolation coefficient was sampled independently at each iteration from the uniform distribution $\mathcal{U}(0, 1)$. For focal loss, we set the focusing parameter to $\gamma = 10$. Table 2 reports the outcomes. Like label smoothing, both Mixup and focal loss markedly reduced merge accuracy relative to the Original configuration, confirming that they also raise prediction entropy and thus interfere with model merging. In every case, however, applying DisTaC restored accuracy to a level on par with Original, demonstrating that DisTaC reliably conditions confidence even when the source models were calibrated with Mixup or focal loss.

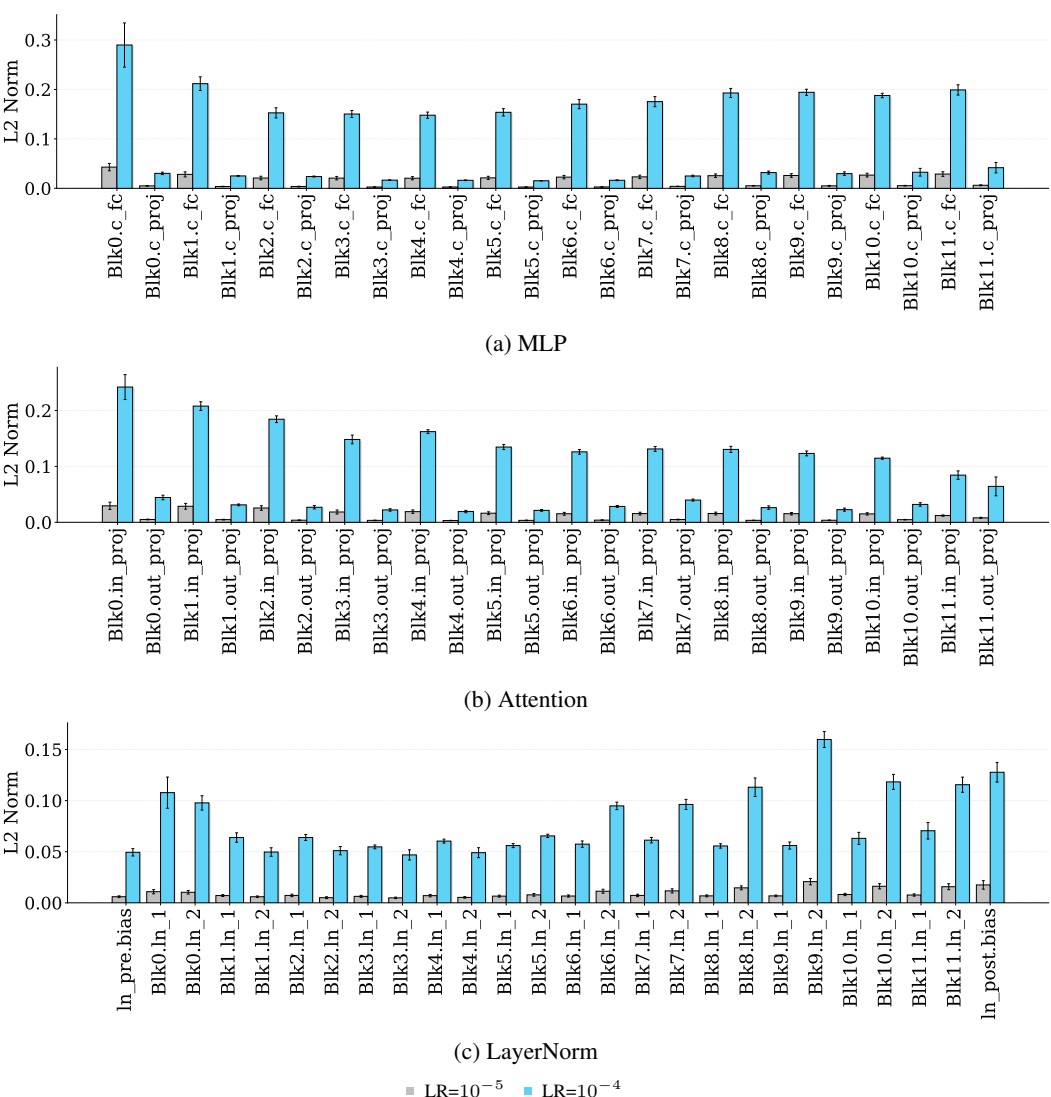

(a) MLP

(b) Attention

(c) LayerNorm

■ LR=$10^{-5}$   ■ LR=$10^{-4}$

Figure 6: **Layer-wise average task-vector norms for bias parameters in ViT-B-32, averaged over eight vision tasks.** Gray bars correspond to a fine-tuning learning rate of $10^{-5}$, blue bars to $10^{-4}$.

Table 2: **Impact of confidence–calibration fine-tuning on merge accuracy.** Source models (ViT-B-32) are fine-tuned with three popular calibration techniques—label smoothing (LS), Mixup, and focal loss—before merging. In every case the resulting merge accuracy drops far below the Original benchmark, showing that low-confidence sources hamper model merging. When the same models are first processed with DisTaC, accuracy is restored to a level on par with Original, confirming that DisTaC's confidence conditioning is effective across all three calibration schemes.

| Method | | Original | LS | Mixup | Focal Loss |
|---|---|---|---|---|---|
| Task arithmetic | | 70.4 (78.0) | 51.0 (58.3) | 52.3 (60.5) | 55.5 (63.9) |
| Task arithmetic | + **DisTac** | - | **63.6** (72.2) | **66.8** (75.2) | **67.2** (76.9) |
| TIES | | 74.0 (82.0) | 54.5 (62.0) | 55.5 (63.9) | 59.4 (68.8) |
| TIES | + **DisTac** | - | **68.7** (77.9) | **69.5** (78.7) | **72.1** (82.4) |
| Consensus TA | | 74.8 (82.8) | 54.6 (62.0) | 54.8 (63.0) | 58.9 (68.2) |
| Consensus TA | + **DisTac** | - | **67.7** (76.5) | **69.4** (77.8) | **71.7** (81.7) |
| TSVM | | 83.3 (92.4) | 60.7 (68.4) | 60.9 (69.6) | 69.3 (79.5) |
| TSVM | + **DisTac** | - | **81.5** (91.8) | **80.1** (90.0) | **81.8** (93.0) |

### E.3 IMPACT OF TASK VECTOR SCALING ON VIT-L-14

We carried out the same scaling experiment (see Figure 3) on the larger ViT-L-14 backbone. As shown in Figure 7, the trend matches that of Figure 3: shrinking the task vector ($\lambda < 1$) leaves single-task accuracy largely unchanged—often even slightly higher—whereas stretching it ($\lambda > 1$) rapidly erodes performance. These results further support the recommendation that, when task-vector norms are mismatched, one should shrink the longer vectors rather than stretch the shorter ones for robust model merging.

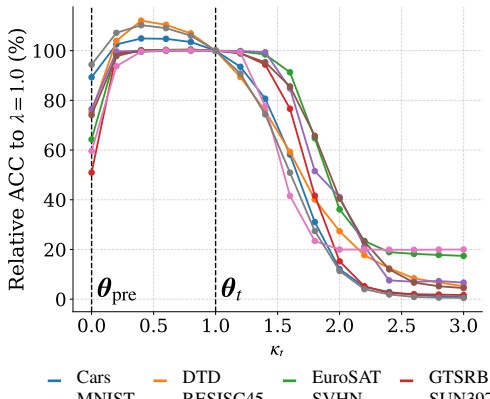

Figure 7: **Effect of scaling task vectors on test accuracy.** For each of the eight vision tasks (ViT-L-14), we evaluate the model $\theta_{\text{pre}} + \lambda\tau$ as the scaling factor $\lambda$ varies from 0.0 to 3.0. Shrinking the task vector ($\lambda < 1.0$) often preserves or even improves accuracy relative to the fine-tuned model ($\lambda = 1.0$), while stretching the vector ($\lambda > 1.0$) leads to sharp degradation. At $\lambda = 3.0$, performance falls below that of the zero-shot model on all tasks. These results support shrinking long task vectors to match shorter ones when resolving norm disparities.

### E.4 SCALING ALONE IS INSUFFICIENT TO OVERCOME NORM MISMATCH

To test whether simple rescaling is sufficient, we revisited the Norm Mismatch scenario and aligned the longest task vector to the mean norm of the remaining vectors before merging. Figure 8 reports the resulting normalized accuracy for ViT B-32 on the eight vision tasks: Original (gray), Norm Mismatch after *only* scaling (light orange), and Norm Mismatch followed by DisTaC (red). The $x$-axis lists the task whose vector was lengthened; "Avg." is the mean over all tasks.

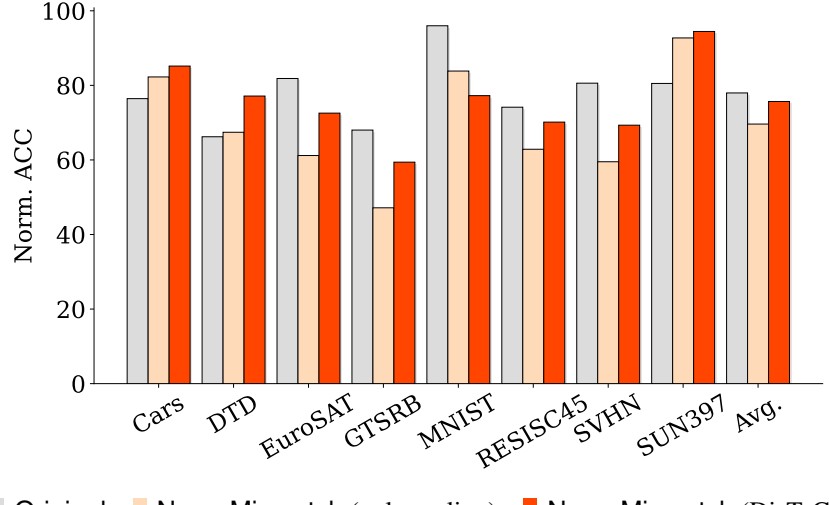

Figure 8: **Normalized merge accuracy for ViT-B-32 on the eight-task benchmark under three conditions.** Gray: Original. Light-orange: Norm Mismatch after rescaling the longest task vector to the mean norm of the others. Red: same rescaled vectors followed by DisTaC. Simple scaling narrows the gap only slightly, whereas DisTaC fully restores accuracy to the Original level. "Avg." denotes the average across all tasks.

Scaling alone lifts accuracy slightly but still leaves a sizeable gap to Original. In contrast, applying DisTaC after scaling recovers the lost performance and matches the baseline across every task. As explained in Section 6.1, even *shrinking* a task vector inevitably hurts its single-task accuracy; DisTaC is therefore essential for restoring that accuracy before merging.

### E.5 SENSITIVITY ANALYSIS ON UNLABELED DATA

We conducted experiments to assess DisTaC's sensitivity to data size and quality.

**Robustness to Data Size.** We first tested DisTaC's performance by varying the number of unlabeled samples per class (100, 200, 300, 400, and 500). Table 3 shows the average relative test accuracy across all tasks, where 100% represents the test accuracy achieved using the full unlabeled dataset (2,490 samples per class on average) for DisTaC. For comparison, we included results for distillation starting directly from the pretrained model ("Distill-from-Pretrained").

The results demonstrate DisTaC's strong robustness to limited data. DisTaC achieves over 90% of the full-data test performance with just 300 samples per class in both failure modes, and maintains over 80% performance even with 100 samples (reaching 96% in the Norm Mismatch case). Compared to distillation from the pretrained model, DisTaC exhibits superior robustness. This highlights the methodological benefit of initializing distillation from the already scaled task vector ($\theta_{pre}+\kappa_t\tau_t$).

Table 3: Relative test accuracy with varying unlabeled data size per class. The baseline (100%) corresponds to test accuracy using the full unlabeled dataset.

| Method | 100 | 200 | 300 | 400 | 500 |
|---|---|---|---|---|---|
| **Norm Mismatch** | | | | | |
| Distill-from-Pretrained | 71.1 | 75.7 | 83.1 | 88.2 | 89.0 |
| DisTaC | **96.0** | **96.0** | **97.3** | **98.6** | **99.0** |
| **Low Confidence** | | | | | |
| Distill-from-Pretrained | 70.1 | 73.8 | 81.2 | 84.6 | 87.6 |
| DisTaC | **83.9** | **87.4** | **90.5** | **91.0** | **95.0** |

**Robustness to Data Quality.** We next assessed robustness to degraded data quality by introducing dataset shift via Gaussian blur during distillation. This setup simulates real-world conditions like variations in weather or camera quality. The blur strength is controlled by the kernel size (fixed at 5) and the intensity range ($\sigma_{\min}, \sigma_{\max}$), where a larger $\sigma$ value indicates stronger corruption. Table 4 shows the relative test accuracy against the performance achieved using clean data for distillation.

The analysis confirms DisTaC's high robustness to quality degradation. DisTaC consistently maintains performance, achieving over $90\%$ of the clean-data performance even under the most severe corruption ($\sigma_{\max} = 3$). In the challenging Low Confidence case, DisTaC maintains near-perfect accuracy (over $98.5\%$) regardless of corruption intensity. DisTaC demonstrates superior robustness compared to the baseline, suggesting that utilizing the original fine-tuned model as the teacher effectively filters noise present in the unlabeled data.

In conclusion, these experiments confirm that DisTaC possesses sufficient robustness to variations in both unlabeled data size and quality, supporting its effectiveness for real-world applications.

Table 4: Relative test accuracy under Gaussian blur corruption. Ranges $[\sigma_{min}, \sigma_{max}]$ denote the blur intensity, with larger values indicating stronger corruption.

| Method | [0.1, 1] | [0.1, 2] | [1, 3] |
|---|---|---|---|
| Norm Mismatch | | | |
| Distill-from-Pretrained | 98.1 | 95.7 | 90.7 |
| DisTaC | **100.4** | **96.2** | **91.7** |
| Low Confidence | | | |
| Distill-from-Pretrained | 98.1 | 97.3 | 94.7 |
| DisTaC | **99.6** | **98.5** | **99.9** |

### E.6 COMPUTATIONAL EFFICIENCY OF DISTAC

Table 5: Computational cost of DisTaC on ViT-B-32 averaged over 8 tasks.

| Metric | Value |
|---|---|
| Hardware | 2 NVIDIA A100 |
| Batch Size | 64 per device |
| Time per Step | $\approx 0.0064$ s |
| Total Time (500 steps) | $\approx 3.2$ s |
| Peak Memory Usage | 7.1 GB |

To empirically validate the claim that DisTaC is computationally lightweight, we measured the training cost using the ViT-B-32 backbone on 2 NVIDIA A100 GPUs. As summarized in Table 5, the distillation process is extremely efficient. With a batch size of 64, the average training time is approximately 0.0064 seconds per step across the eight vision tasks. Consequently, the standard 500-step DisTaC procedure requires only about 3.2 seconds to complete (excluding evaluation time). The peak GPU memory usage was recorded at 7.1 GB, which includes the overhead for online teacher inference; this could be further optimized by pre-computing teacher predictions.

### E.7 GENERALIZING DISTAC TO NLP

We conducted experiments using RoBERTa-base (RoBERTa-b), RoBERTa-large (RoBERTa-l) (Zhuang et al., 2021), and Llama2-7b (Touvron et al., 2023) to examine whether our claims extend beyond vision tasks to the NLP domain. Following Ilharco et al. (2023), we adopt four GLUE benchmark(Wang et al., 2019) tasks: CoLA, MRPC, RTE, and SST-2. In the NLP experiments, we evaluate the same settings as in vision: Norm Mismatch and Low Confidence.

The results are presented in Table 6. In comparison to the original configuration, the normalized score degrades under both Norm Mismatch and Low Confidence settings. In instances of norm mismatch among task vectors, the application of DisTaC effectively reduces interference between task vectors, thereby enhancing the normalized score from that of task arithmetic without DisTaC

Table 6: **Comparison of post-merge accuracy across fine-tuning configurations and the effect of DisTaC in NLP.** Absolute accuracy is displayed in a large font size, whereas normalized accuracy appears in parentheses in a smaller font. When the task vector norms diverge (Norm Mismatch) or the source models exhibit low confidence (Low Confidence), the normalized score degrades relative to the standard benchmark setting (Original). Under these conditions, DisTaC effectively pre-conditions the source models, improving performance in both settings.

| Method | | Original | | | Norm Mismatch | | | Low Confidence | | |
|---|---|---|---|---|---|---|---|---|---|---|
| | | RoBERTa-b | RoBERTa-l | Llama2-7b | RoBERTa-b | RoBERTa-l | Llama2-7b | RoBERTa-b | RoBERTa-l | Llama2-7b |
| Task arithmetic | | 60.9 (73.5) | 68.3 (82.4) | 75.9 (91.7) | 56.8 (68.5) | 46.0 (58.1) | 55.3 (64.7) | 61.3 (72.6) | 64.5 (73.9) | 75.7 (95.1) |
| Task arithmetic | + DisTaC | - | - | - | **59.9** (71.7) | **64.4** (80.5) | **75.0** (91.1) | **62.5** (74.6) | **70.0** (82.3) | 73.0 (95.9) |
| Ties-merging | | 60.9 (74.8) | 65.7 (80.7) | 58.3 (80.7) | 39.9 (46.1) | 40.8 (51.3) | 40.6 (47.7) | 65.4 (79.1) | 71.8 (84.0) | 38.3 (47.5) |
| Ties-merging | + DisTaC | - | - | - | **62.4** (76.4) | **59.4** (75.9) | **44.0** (51.6) | 64.4 (78.0) | **72.5** (86.4) | **58.9** (78.4) |
| TSVM | | 65.8 (80.8) | 72.0 (87.8) | 66.1 (78.5) | 58.8 (71.1) | 48.0 (60.7) | 55.5 (65.5) | 69.6 (84.3) | 73.3 (85.6) | 68.1 (84.6) |
| TSVM | + DisTaC | - | - | - | **65.1** (79.6) | **66.3** (84.1) | **64.6** (77.4) | 67.5 (82.4) | **75.8** (90.8) | **72.4** (97.1) |
| Consensus-merging | | 61.3 (73.7) | 67.9 (81.4) | 74.5 (89.7) | 58.1 (70.0) | 38.1 (47.3) | 58.3 (68.6) | 61.2 (72.3) | 65.2 (75.5) | 65.0 (79.1) |
| Consensus-merging | + DisTaC | - | - | - | **60.5** (72.5) | **63.4** (79.0) | **68.4** (82.3) | **62.2** (74.3) | **69.8** (82.3) | **72.0** (94.9) |

(e.g., RoBERTa-large exhibits an increase from $58.1$ to $80.5$, an improvement of $22.4$ points in the normalized score). Furthermore, when the task vectors exhibit low confidence, the implementation of DisTaC results in an elevation of the normalized score compared to scenarios without DisTaC (e.g., RoBERTa-large: $73.9$ to $82.3$, an enhancement of $8.4$ points in the normalized score). These findings indicate that (i) the identified failure modes of norm disparity and low confidence we identify arise in both vision and language tasks, and (ii) DisTaC conditioning consistently enhances the outcome of merging for CLIP/ViT, Roberta, and Llama. We posit that these results demonstrate the cross-modality generalizability of vision and language. Notably, the recovery is stronger at larger scales (e.g., llama2-7b in Norm Mismatch), suggesting that the method retains its efficacy as model capacity expands.

