# OpenReview forum: "DisTaC: Conditioning Task Vectors via Distillation for Robust Model Merging"
_ICLR.cc/2026/Conference — ICLR 2026 Poster_

### Official Review · Reviewer_bqXy · 2025-10-20

**Soundness:** 3
**Presentation:** 3
**Contribution:** 3
**Rating:** 6
**Confidence:** 3

**Summary:**

The authors identify two critical failure modes that are common in practical scenarios but overlooked in idealized benchmarks: (1) Task Vector Norm Disparity, where task vectors have vastly different magnitudes; and (2) Low-Confidence Source Models, where models produce high-entropy predictions, often as a result of calibration techniques like label smoothing.

The paper proposes a lightweight pre-conditioning method. DisTaC uses KD on unlabeled data to prepare models for merging. To correct norm disparities, it first rescales a task vector to a target norm and then uses KD to distill knowledge from the original, high-performing model, thereby recovering the performance lost during scaling. To correct for low confidence, it distills knowledge using a higher temperature for the student model than the teacher, which encourages the student to produce lower-entropy, more confident predictions.

The authors demonstrate empirically on eight vision tasks with ViT-B-32 and ViT-L-14 backbones that these failure modes significantly degrade the performance of state-of-the-art merging methods. They then show that applying DisTaC before merging consistently restores performance to the level of the idealized benchmark, with minimal computational overhead. Finally, the paper offers practical guidelines for model merging.

**Strengths:**

1. The paper's primary contribution is the identification and diagnosis of two practical failure modes in model merging. While the component techniques (knowledge distillation) are not new, their application to pre-condition task vectors to solve these specific robustness issues is novel.

2. The claims are substantiated by a thorough and well-designed set of experiments. The inclusion of theoretical motivation (Proposition 1 and the analysis in Appendix C) adds rigor and provides a deeper understanding of the observed phenomena. The ablation studies, such as the analysis of shrinking versus stretching vectors in Figure 3, are particularly convincing and support the practical guidelines offered.

3. By addressing the gap between idealized academic benchmarks, the paper provides a crucial step towards making model merging a more practical and reliable tool. DisTaC is a simple, effective, and computationally cheap solution that can be readily adopted by practitioners.

**Weaknesses:**

* The experiments are confined exclusively to CLIP-based models on vision tasks. The general applicability of DisTaC would be strengthened by demonstrating its effectiveness in other modalities, such as merging fine-tuned LLMs, or on MLLMs.

* The method's reliance on unlabeled data from the task distribution is a potential limitation. The paper would benefit from an analysis of the method's sensitivity to the quantity of this unlabeled data. For instance, how much data is needed for DisTaC to be effective, and how does performance degrade if the unlabeled data is from a slightly shifted distribution?

**Questions:**

1. Figure 3 and the discussion in Section 6.1 suggest that shrinking a task vector can sometimes lead to better performance than the original model. This is a very interesting observation. Could you elaborate on the potential reasons for this? Does this imply a regularization effect, suggesting that the initial fine-tuning may have overshot a more optimal, shorter task vector?

2. Regarding the influence of learning rate and training steps on the effectiveness of model merging, you may refer to the error analysis in Theorem 3.1 of [1], where the impact of the norm of the task vector was also observed. In [2], analysis of the properties of task vectors was conducted.

*[1] Unifying Multimodal Large Language Model Capabilities and Modalities via Model Merging. arXiv preprint arXiv:2505.19892*
*[2] MAP: Low-compute Model Merging with Amortized Pareto Fronts via Quadratic Approximation. arXiv preprint arXiv:2406.07529*

---

> ### Author Response · Authors · 2025-11-21
> **Response to reviewer bqXy**
>
> # Regarding weakness 1
>
> We appreciate your concern that our evaluation initially focused on CLIP/ViT models for vision tasks and did not include language tasks. We conducted additional experiments on RoBERTa-base and RoBERTa-large across four tasks from GLUE benchmark [1] to test the same failure modes (norm disparity and low confidence) and the effectiveness of DisTaC.
>
> | Model: roberta-base      |   Original  | Norm Mismatch | Low Confidence |
> |--------------------------|-------------|---------------|----------------|
> | Task arithmetic          | 60.9 (73.5) |  56.8 (68.5)  |   61.3 (72.6)  |
> | Task arithmetic + DisTaC |      -      |  **59.9** (**71.7**)  |   **62.5** (**74.6**)  |
>
> | Model: roberta-large     |   Original  | Norm Mismatch | Low Confidence |
> |--------------------------|-------------|---------------|----------------|
> | Task arithmetic          | 68.3 (82.4) |  46.0 (58.1)  |   64.5 (73.9)  |
> | Task arithmetic + DisTaC |      -      |  **64.4** (**80.5**)  |   **70.0** (**82.3**)  |
>
> In the two tables above, we showed experimental results with RoBERTa-base and RoBERTa-large. Absolute score is displayed outside of parentheses, whereas normalized score appears in parentheses. The score in the table is based on the average of evaluation metrics defined in GLUE benchmark. Compared to the Original setting, the normalized score degrades under both Norm Mismatch and Low Confidence settings. When there is norm mismatch between task vectors, applying DisTaC mitigates the interference between task vectors so that the normalized score is improved from the one of task arithmetic without DisTaC (e.g., RoBERTa-large: 58.1->80.5, +22.4 points in the normalized score). When the task vectors have low confidence, applying DisTaC improves the normalized score from the one without DisTaC (e.g., RoBERTa-large: 73.9->82.3, +8.4 points in the normalized score).
>
> These results demonstrate that (i) the two failure modes of norm disparity and low confidence we identify arise both in vision tasks and language tasks and (ii) DisTaC conditioning consistently improves merging outcome for CLIP/ViT and RoBERTa models. We believe these findings show modality generalizability of vision and language. Importantly, the recovery is stronger at larger scale (roberta-large), suggesting the approach remains useful as model capacity grows.
>
> We will incorporate these results and other merging methods (e.g., TIES-merging, Consensus merging, and TSVM) in the camera-ready version and revise the Introduction and Limitations to clarify the scope, generality, and modality positioning.
>
> [1] Wang, Alex, et al. "GLUE: A multi-task benchmark and analysis platform for natural language understanding." Proceedings of the 2018 EMNLP workshop BlackboxNLP: Analyzing and interpreting neural networks for NLP. 2018.

---

> ### Author Response · Authors · 2025-11-21
> **Response to reviewer bqXy**
>
> # Regarding weakness 2
>
> We appreciate your comment on the sensitivity of DisTaC to the amount of unlabeled data.
> We acknowledge the need for a sensitivity analysis regarding the quality of unlabeled data to fully substantiate DisTaC's robustness claims. We conducted experiments examining the effects of both unlabeled data size and data quality on DisTaC's performance.
>
> We first tested DisTaC's performance by varying the number of unlabeled samples per class (100, 200, 300, 400, and 500). The table below shows the average relative test accuracy across all tasks, where $100$ represents the test accuracy achieved using the full unlabeled dataset for DisTaC. For comparison, we included results for distillation starting directly from the pretrained model ("Distill-from-Pretrained").
>
> || 100  | 200  | 300  | 400  | 500  |
> |-----------|------|------|------|------|------|
> | [Norm Mismatch]        |   |   |   |   |   |
> | Distill-from-Pretrained  | 71.1 | 75.7 | 83.1 | 88.2 | 89.0 |
> | DisTaC   | **96.0** | **96.0** | **97.3** | **98.6** | **99.0** |
> | [Low Confidence]        |   |   |   |   |   |
> | Distill-from-Pretrained      | 70.1  | 73.8  | 81.2  | 84.6  | 87.6  |
> | DisTaC     | **83.9**  | **87.4**  | **90.5**  | **91.0**  | **95.0**  |
>
> The results demonstrate DisTaC's strong robustness to limited data:
> - DisTaC achieves over $90\%$ of the full-data test performance with just $\mathbf{300}$ samples per class in both failure modes, and maintains over $80\%$ performance even with $\mathbf{100}$ samples (reaching $96\%$ in the Norm Mismatch case).
> - Compared to distillation from the pretrained model, DisTaC exhibits superior robustness. This highlights the methodological benefit of initializing distillation from the already scaled task vector ($\theta_{pre} + \kappa_t \tau_t$).
>
>
>
> We next assessed robustness to degraded data quality by introducing dataset shift via Gaussian blur during distillation. This setup simulates real-world conditions like variations in weather or camera quality. The blur strength is controlled by the kernel size (fixed at 5) and the intensity range ($\sigma_{\min}, \sigma_{\max}$), where a larger $\sigma$ value indicates stronger corruption.
> The results below show the relative test accuracy against the performance achieved using clean data for distillation.
>
> |    | [0.1, 1] | [0.1, 2] | [1, 3] |
> |-----------|----------|----------|--------|
> | [Norm Mismatch]        |   |   |   |   |   |
> | Distill-from-Pretrained  | 98.1     | 95.7     | 90.7   |
> | DisTaC  | **100.4**    | **96.2**     | **91.7**   |
> | [Low Confidence]        |   |   |   |   |   |
> | Distill-from-Pretrained  | 98.1     | 97.3     | 94.7   |
> | DisTaC  | **99.6**    | **98.5**     | **99.9**   |
>
>
> The analysis confirms DisTaC's high robustness to quality degradation:
> - DisTaC consistently maintains performance, achieving over $90\%$ of the clean-data performance even under the most severe corruption ($\sigma_{\max}=3$).
> - In the challenging Low Confidence case, DisTaC maintains near-perfect accuracy (over $98.5\%$) regardless of corruption intensity.
> - DisTaC demonstrates superior robustness compared to the baseline, suggesting that utilizing the original fine-tuned model as the teacher effectively filters noise present in the unlabeled data.
>
> **In conclusion, these experiments confirm that DisTaC possesses sufficient robustness to variations in both unlabeled data size and quality, supporting its effectiveness for real-world application**.
>
> We have appended the above discussion to the paper’s appendix for completeness.

---

> ### Author Response · Authors · 2025-11-21
> **Response to reviewer bqXy**
>
> # Regarding question 1
>
> We thank you for your deep and insightful question regarding shrinking a task vector.
>
> We fully agree with your interpretation. We observed that for tasks with limited training data, such as Cars and DTD, fine-tuning tends to overfit the training set. Consequently, shrinking the task vector acts as a form of pseudo-regularization, effectively correcting this overshoot and improving generalization performance. Theoretically, this can be rigorously explained within the Neural Tangent Kernel (NTK) regime, where shrinking the task vector is equivalent to early stopping. In the NTK regime, the relationship between task vector scaling and model output is approximated linearly as:
> $$f(x;\theta_0 + \alpha \tau) \approx f(x;\theta_0) + \alpha\tau^{\top}\nabla_\theta f(x;\theta_0)$$
> Under this linearization, if we assume a convex loss function, neural network training reduces to convex optimization, and the optimization trajectory from the initialization to the optimum becomes linear—coinciding with the direction of the task vector. Therefore, applying a scalar $\alpha < 1$ to the trained task vector is geometrically equivalent to halting the training process earlier along this trajectory (early stopping). This theoretical insight supports our empirical finding that merging "shorter" vectors (which benefit from this implicit regularization) yields better performance than merging "longer" vectors, reinforcing our guideline to resolve norm disparities by shrinking larger vectors rather than stretching smaller ones.

---

> ### Author Response · Authors · 2025-11-21
> **Response to reviewer bqXy**
>
> # Regarding question 2
>
> We thank the reviewer for bringing these valuable references to our attention.
> We agree that Theorem 3.1 in [1], which analyzes error bounds with respect to the task vector norm, provides a crucial theoretical foundation for our empirical observations regarding learning rates and training steps. Specifically, it theoretically corroborates our finding in Section 6.1 and Figure 3 that "shrinking" task vectors (reducing their norm) effectively keeps the model within the local linear regime where merging is most effective. Similarly, the analysis of task vector properties in [2] offers important context for our work. We will incorporate these citations into our Related Work and Discussion (Section 6.1) sections to strengthen the theoretical justification for why addressing norm disparity is essential for robust model merging.

---

> > ### Comment · Reviewer_bqXy · 2025-11-27
> >
> > The rebuttal substantially improves my understanding of the work and addresses my main technical concerns, especially regarding robustness to unlabeled data. I would like to keep my positive score.
> > By the way, I encourage the authors to clearly position the current work as being primarily validated on vision models and encoder-style language models (RoBERTa). These architectures differ from modern decoder-style LLMs, so it would be helpful to discuss extending DisTaC to contemporary LLMs as future work.

---

> > > ### Author Response · Authors · 2025-11-27
> > > **Reply to reviewer bqXy**
> > >
> > > We are pleased to hear that our rebuttal resolved your concern.
> > >
> > > Regarding LLMs, as other reviewers also suggested, we have already conducted additional experiments using Llama2‑7B. We summarize the results below and will integrate full details into the updated manuscript.
> > >
> > > |                        |   Original  | Norm Mismatch | Low Confidence |
> > > |------------------------|-------------|---------------|----------------|
> > > | task addition          | 75.9 (91.7) |  55.3 (64.7)  |   75.7 (95.1)  |
> > > | task addition + DisTaC |      -      |  **75.0** (**91.1**)  |   74.8 (**98.5**)  |
> > >
> > > Key Findings:
> > > - Norm Mismatch: DisTaC provides a substantial recovery, improving the absolute score by nearly **20 points (55.3 → 75.0)** and recovering the normalized score to **91.1**, which is comparable to the Original setting.
> > > - Low Confidence: **DisTaC improves the normalized score (98.5 vs. 95.1)**, indicating that the merged model acts more similarly to the ideal single-task models, while maintaining comparable absolute performance to the baseline. These results on a 7B-parameter model confirm that DisTaC is highly effective for LLMs, scaling well beyond the RoBERTa and ViT architectures.
> > >
> > > We have added these Llama-2-7B results to the Appendix.
> > >
> > > We sincerely hope that these additional experiments have fully addressed all of your concerns. If so, we would respectfully request that you reconsider your score.

---

### Official Review · Reviewer_P2H4 · 2025-10-22

**Soundness:** 3
**Presentation:** 3
**Contribution:** 3
**Rating:** 6
**Confidence:** 2

**Summary:**

The paper investigates the robustness of model merging methods. The authors identify two critical failure modes in existing model-merging pipelines:

+ Task vector norm disparities — differences in magnitudes caused by heterogeneous fine-tuning hyperparameters (e.g., learning rates, steps).

+ Low pretrained-model confidence — often resulting from training techniques such as label smoothing, Mixup, or focal loss.

To mitigate these issues, the authors propose DisTaC, a lightweight rescaling task vectors to a target norm and increasing model confidence by training students with higher temperatures than teachers. Empirical evaluations on eight vision tasks demonstrate that DisTaC restores and often improves post-merge accuracy.

**Strengths:**

The experiments are extensive, including ablation studies, visual analyses (e.g., layer-wise norm shifts in Figs. 5–6), and additional tests with Mixup and focal loss to demonstrate generality.

The authors go beyond idealized benchmarks and pinpoint practical causes of model-merging failures, which is interesting and meaningful.

**Weaknesses:**

The experiments are restricted to vision-only tasks using CLIP ViT backbones. No tests are conducted on NLP, speech, or multimodal architectures.

While the paper argues that unlabeled data are “readily available,” this assumption may not hold in all domains, such as the security issue, or we can not find a comprehensive set for LLMs.

While DisTaC generally improves results, the paper lacks a sensitivity analysis of unlabeled data quality that would clarify when it might fail or overfit.

The authors assert that DisTaC is computationally lightweight (“500 steps, unlabeled data only”), but do not quantitatively compare its runtime or cost to baseline merges, nor provide results on large-scale LLMs such as Qwen.

**Questions:**

Why not provide the results of the Original version when combined with DisTaC in Table 1? It is helpful for us to understand that DisTaC may be harmful for performance in some cases.

---

> ### Author Response · Authors · 2025-11-21
> **Response to reviewer P2H4**
>
> # Regarding weakness 1
>
> We appreciate your concern that our evaluation initially focused on CLIP/ViT models for vision tasks and did not include language tasks. We conducted additional experiments on RoBERTa-base and RoBERTa-large across four tasks from GLUE benchmark [1] to test the same failure modes (norm disparity and low confidence) and the effectiveness of DisTaC.
>
> | Model: roberta-base      |   Original  | Norm Mismatch | Low Confidence |
> |--------------------------|-------------|---------------|----------------|
> | Task arithmetic          | 60.9 (73.5) |  56.8 (68.5)  |   61.3 (72.6)  |
> | Task arithmetic + DisTaC |      -      |  **59.9** (**71.7**)  |   **62.5** (**74.6**)  |
>
> | Model: roberta-large     |   Original  | Norm Mismatch | Low Confidence |
> |--------------------------|-------------|---------------|----------------|
> | Task arithmetic          | 68.3 (82.4) |  46.0 (58.1)  |   64.5 (73.9)  |
> | Task arithmetic + DisTaC |      -      |  **64.4** (**80.5**)  |   **70.0** (**82.3**)  |
>
> In the two tables above, we showed experimental results with RoBERTa-base and RoBERTa-large. Absolute score is displayed outside of parentheses, whereas normalized score appears in parentheses. The score in the table is based on the average of evaluation metrics defined in GLUE benchmark. Compared to the Original setting, the normalized score degrades under both Norm Mismatch and Low Confidence settings. When there is norm mismatch between task vectors, applying DisTaC mitigates the interference between task vectors so that the normalized score is improved from the one of task arithmetic without DisTaC (e.g., RoBERTa-large: 58.1->80.5, +22.4 points in the normalized score). When the task vectors have low confidence, applying DisTaC improves the normalized score from the one without DisTaC (e.g., RoBERTa-large: 73.9->82.3, +8.4 points in the normalized score).
>
> These results demonstrate that (i) the two failure modes of norm disparity and low confidence we identify arise both in vision tasks and language tasks and (ii) DisTaC conditioning consistently improves merging outcome for CLIP/ViT and RoBERTa models. We believe these findings show modality generalizability of vision and language. Importantly, the recovery is stronger at larger scale (roberta-large), suggesting the approach remains useful as model capacity grows.
>
> We will incorporate these results and other merging methods (e.g., TIES-merging, Consensus merging, and TSVM) in the camera-ready version and revise the Introduction and Limitations to clarify the scope, generality, and modality positioning.
>
> [1] Wang, Alex, et al. "GLUE: A multi-task benchmark and analysis platform for natural language understanding." Proceedings of the 2018 EMNLP workshop BlackboxNLP: Analyzing and interpreting neural networks for NLP. 2018.

---

> ### Author Response · Authors · 2025-11-21
> **Response to reviewer P2H4**
>
> # Regarding weakness 2
>
> We sincerely appreciate your valuable insights regarding access to unlabeled data.
>
> We acknowledge that DisTaC's reliance on unlabeled data may pose limitations in certain specialized domains, such as those with stringent security requirements or complex data distributions like a comprehensive set for LLMs. While a fully data-free approach to addressing all failure modes remains the ideal, we assert that DisTaC represents the first systematic attempt to identify the specific failure modes (norm disparity and low confidence) that critically undermine state-of-the-art merging methods and to propose a practical, effective remedy.
>
> Current state-of-the-art merging techniques, including TSVM and others, are indeed largely data-free aside from coefficient tuning, and perform well under idealized benchmark conditions. Yet, they suffer significant performance degradation under the realistic failure modes we introduced. We contend that the use of unlabeled data in DisTaC is currently the most potent and realistic auxiliary means to restore robustness in these challenging scenarios.
>
> **Furthermore, we demonstrated DisTaC's practical viability by confirming its strong robustness to variations in both data size and quality as detailed in our subsequent response. This robust performance suggests that DisTaC remains effective even in real-world scenarios where sufficient high-quality unlabeled data may be difficult to gather**.
>
> We agree that achieving a complete data-free solution for these identified failure modes is extremely challenging and remains an important direction for future work.

---

> ### Author Response · Authors · 2025-11-21
> **Response to reviewer P2H4**
>
> # Regarding weakness 3
>
> Thank you for your concern regarding DisTaC's sensitivity to the size and quality of unlabeled data.
>
> We acknowledge the need for a sensitivity analysis regarding the quality of unlabeled data to fully substantiate DisTaC's robustness claims. We conducted experiments examining the effects of both unlabeled data size and data quality on DisTaC's performance.
>
> We first tested DisTaC's performance by varying the number of unlabeled samples per class (100, 200, 300, 400, and 500). The table below shows the average relative test accuracy across all tasks, where $100\%$ represents the test accuracy achieved using the full unlabeled dataset (2490 samples per class on average) for DisTaC. For comparison, we included results for distillation starting directly from the pretrained model ("Distill-from-Pretrained").
>
> || 100  | 200  | 300  | 400  | 500  |
> |-----------|------|------|------|------|------|
> | [Norm Mismatch]        |   |   |   |   |   |
> | Distill-from-Pretrained  | 71.1 | 75.7 | 83.1 | 88.2 | 89.0 |
> | DisTaC   | **96.0** | **96.0** | **97.3** | **98.6** | **99.0** |
> | [Low Confidence]        |   |   |   |   |   |
> | Distill-from-Pretrained      | 70.1  | 73.8  | 81.2  | 84.6  | 87.6  |
> | DisTaC     | **83.9**  | **87.4**  | **90.5**  | **91.0**  | **95.0**  |
>
> The results demonstrate DisTaC's strong robustness to limited data:
> - DisTaC achieves over $90\%$ of the full-data test performance with just $\mathbf{300}$ samples per class in both failure modes, and maintains over $80\%$ performance even with $\mathbf{100}$ samples (reaching $96\%$ in the Norm Mismatch case).
> - Compared to distillation from the pretrained model, DisTaC exhibits superior robustness. This highlights the methodological benefit of initializing distillation from the already scaled task vector ($\theta_{pre} + \kappa_t \tau_t$).
>
> We next assessed robustness to degraded data quality by introducing dataset shift via Gaussian blur during distillation. This setup simulates real-world conditions like variations in weather or camera quality. The blur strength is controlled by the kernel size (fixed at 5) and the intensity range ($\sigma_{\min}, \sigma_{\max}$), where a larger $\sigma$ value indicates stronger corruption.
> The results below show the relative test accuracy against the performance achieved using clean data for distillation.
>
> |    | [0.1, 1] | [0.1, 2] | [1, 3] |
> |-----------|----------|----------|--------|
> | [Norm Mismatch]        |   |   |   |   |   |
> | Distill-from-Pretrained  | 98.1     | 95.7     | 90.7   |
> | DisTaC  | **100.4**    | **96.2**     | **91.7**   |
> | [Low Confidence]        |   |   |   |   |   |
> | Distill-from-Pretrained  | 98.1     | 97.3     | 94.7   |
> | DisTaC  | **99.6**    | **98.5**     | **99.9**   |
>
>
> The analysis confirms DisTaC's high robustness to quality degradation:
> - DisTaC consistently maintains performance, achieving over $90\%$ of the clean-data performance even under the most severe corruption ($\sigma_{\max}=3$).
> - In the challenging Low Confidence case, DisTaC maintains near-perfect accuracy (over $98.5\%$) regardless of corruption intensity.
> - DisTaC demonstrates superior robustness compared to the baseline, suggesting that utilizing the original fine-tuned model as the teacher effectively filters noise present in the unlabeled data.
>
> In conclusion, these experiments confirm that DisTaC possesses sufficient robustness to variations in both unlabeled data size and quality, supporting its effectiveness for real-world application.
>
> We have appended the above discussion to the paper’s appendix for completeness.

---

> ### Author Response · Authors · 2025-11-21
> **Response to reviewer P2H4**
>
> # Regarding weakness 4
>
> We thank the reviewer for the constructive comment regarding the computational efficiency and scalability of DisTaC.
>
> We agree that quantitative runtime metrics are necessary to substantiate the claim that DisTaC is computationally lightweight. We have measured the computational cost for the ViT-B-32 backbone used in the experiments for Table 1.
> Averaging across the eight tasks, the training time per step was approximately $0.0064$ seconds on 2 NVIDIA A100 GPUs (batch size 64). Consequently, the total time required to run the full 500 DisTaC steps is calculated to be around 3.2 seconds (excluding the time for checkpoint evaluation). The peak GPU memory usage recorded during this process was 7.1 GB. We also note that this measurement includes online inference of the teacher model during distillation; if the teacher's predictions were pre-computed, memory usage could be further optimized. This quantitative data clearly demonstrates that DisTaC imposes only negligible computational overhead on the overall model merging pipeline.
> We have added these quantitative runtime results to the paper’s appendix.
>
> Regarding results on LLMs, as detailed in our response to Weaknesses 1, we conducted experiments on RoBERTa‑base/large for NLP tasks and confirmed DisTaC’s effectiveness, mirroring the findings on vision tasks.

---

> ### Author Response · Authors · 2025-11-21
> **Response to reviewer P2H4**
>
> # Regarding question 1
>
> We thank you for your insightful question regarding the "Original + DisTaC" setting.
>
> Our primary reason for omitting the "Original + DisTaC" results was conceptual. DisTaC is fundamentally designed as a diagnostic-driven intervention for models exhibiting specific pathologies: norm disparity or low confidence. The "Original" configuration is explicitly defined as the ideal benchmark where task vectors are healthy (uniform learning rate, hard labels, uniform norm). There is no technical motivation to apply our corrective conditioning step in this scenario.
>
> However, applying DisTaC to a healthy task vector would essentially reduce the process to Self-Distillation, since the scaling factor $\kappa_t$ would be approximately $1.0$ (due to uniform norms) and the target confidence is already high. We argue that this is unlikely to cause significant degradation, as self-distillation is generally known to maintain or slightly improve model performance[2][3].
>
> We acknowledge the necessity of empirically proving this safety claim. We are currently running the experiment combining DisTaC with the "Original" configuration and plan to include these results in the camera-ready version of the paper for complete transparency.
>
> [2] Zhang, Linfeng, et al. "Be your own teacher: Improve the performance of convolutional neural networks via self distillation." Proceedings of the IEEE/CVF international conference on computer vision. 2019.
>
> [3] Pareek, Divyansh, Simon S. Du, and Sewoong Oh. "Understanding the gains from repeated self-distillation." Advances in Neural Information Processing Systems 37 (2024): 7759-7796.

---

> > ### Comment · Reviewer_P2H4 · 2025-11-27
> >
> > Thanks for the detailed reply. Most of my concerns are addressed. I would like to keep my positive score.
> > By the way, I think the words "readily available" should be modified, since it is not rigorous. I do not doubt this assumption since many methods use a few exemplars during merging.

---

> ### Author Response · Authors · 2025-11-27
> **Reply to reviewer P2H4**
>
> We are glad that we have addressed most of your concerns.
>
> Thank you for your thoughtful feedback regarding the phrasing about access to unlabeled data. We agree that “readily available” is insufficiently rigorous. We have revised the manuscript to explicitly qualify domain dependence and limitations on data availability, and updated the corresponding section on limitations to reflect these constraints and the robustness analysis under limited quantity and mild distribution shift.
>
> > Additionally, DisTaC assumes access to unlabeled data for distillation, which can at times be challenging due to potential security constraints. Nevertheless, we emphasize that DisTaC achieves over 96\% of ideal performance even when using extremely small datasets or data with severe distribution shifts, demonstrating strong robustness in such settings (see Appendix C.5).
>
> We believe these revisions and additional experiments fully address all of your concerns, and we would respectfully request that you reconsider your score.

---

### Official Review · Reviewer_XSii · 2025-10-31

**Soundness:** 2
**Presentation:** 3
**Contribution:** 2
**Rating:** 4
**Confidence:** 4

**Summary:**

The paper investigates the robustness limits of state-of-the-art multi-task model merging methods, identifying two critical failure modes: disparities in task vector norms and low confidence in source models. To address these, the authors propose DisTaC—a pre-conditioning framework utilizing knowledge distillation to harmonize task vector norms and boost model confidence before merging. Extensive experiments on eight vision tasks with CLIP backbones demonstrate that DisTaC consistently restores or enhances merging performance under challenging, realistic scenarios where conventional merging fails.

**Strengths:**

1. The paper provides a sharp empirical and theoretical analysis of robustness shortcomings in existing model merging approaches, uncovering how norm disparity and low confidence can produce significant accuracy degradation after merging.
2. DisTaC is a clear, well-motivated conditioning method, leveraging knowledge distillation with unlabeled data for practical pre-conditioning. By integrating both norm scaling and confidence sharpening, it directly addresses the two key failure modes.

**Weaknesses:**

1. Since the pre-conditioning step is applied independently to each source model, the total computational cost scales linearly with the number of models. For large-scale ensembles, this cumulative overhead can become prohibitive, undermining the method's primary efficiency advantage over full-scale retraining. The distillation cost also scales with model parameter count, becoming computationally prohibitive for large-scale foundation models (e.g., LLMs).
2. The study’s empirical validation is confined to vision tasks, leaving its applicability to other modalities, particularly Large Language Models (LLMs), unverified. Given that task vector dynamics in LLMs may be substantially more complex, the method's effectiveness for merging models with diverse reasoning and generative capabilities is not guaranteed.

**Questions:**

1. How does DisTaC's independent conditioning of each task vector affect the final merge performance when tasks are fundamentally antagonistic or highly complementary?

---

> ### Author Response · Authors · 2025-11-21
> **Response to reviewer XSii**
>
> # Regarding weakness 1
>
> We appreciate your insightful feedback regarding the scalability of DisTaC.
>
> We acknowledge the reviewer's concern regarding the potential cumulative overhead of applying DisTaC to a large number of models and the scalability challenges with LLMs.
>
> Firstly, to address the cost per model, we strongly maintain that DisTaC introduces only negligible overhead compared to the original fine-tuning process or full-scale multi-task retraining. Our experiments using the ViT-B-32 backbone show that running the full 500 DisTaC steps requires only approximately 3.2 seconds per task on 2 NVIDIA A100 GPUs (0.0064 sec per step). The cost is trivial, and even scaling linearly, the total time for dozens of models would likely remain far quicker than retraining a single shared model from scratch.
>
> Secondly, and more importantly, it is not necessary to apply DisTaC to every task vector. DisTaC is a diagnostic-driven intervention. We only propose pre-conditioning a model if its source characteristics$-$task vector norm or prediction confidence$-$are identified as detrimental to the merging process. A user should first diagnose the input task vectors (e.g., check for large norm disparities as shown in Figure 1b or low confidence/high entropy as shown in Figure 1c) and apply DisTaC selectively to only the minority of models that exhibit these harmful traits, drastically reducing the overall cumulative cost and preserving the method's efficiency advantage.

---

> ### Author Response · Authors · 2025-11-21
> **Response to reviewer XSii**
>
> # Regarding weakness 2
>
> We appreciate your concern that our evaluation initially focused on CLIP/ViT models for vision tasks and did not include language tasks. We conducted additional experiments on RoBERTa-base and RoBERTa-large across four tasks from GLUE benchmark [1] to test the same failure modes (norm disparity and low confidence) and the effectiveness of DisTaC.
>
> | Model: roberta-base      |   Original  | Norm Mismatch | Low Confidence |
> |--------------------------|-------------|---------------|----------------|
> | Task arithmetic          | 60.9 (73.5) |  56.8 (68.5)  |   61.3 (72.6)  |
> | Task arithmetic + DisTaC |      -      |  **59.9** (**71.7**)  |   **62.5** (**74.6**)  |
>
> | Model: roberta-large     |   Original  | Norm Mismatch | Low Confidence |
> |--------------------------|-------------|---------------|----------------|
> | Task arithmetic          | 68.3 (82.4) |  46.0 (58.1)  |   64.5 (73.9)  |
> | Task arithmetic + DisTaC |      -      |  **64.4** (**80.5**)  |   **70.0** (**82.3**)  |
>
> In the two tables above, we showed experimental results with RoBERTa-base and RoBERTa-large. Absolute score is displayed outside of parentheses, whereas normalized score appears in parentheses. The score in the table is based on the average of evaluation metrics defined in GLUE benchmark. Compared to the Original setting, the normalized score degrades under both Norm Mismatch and Low Confidence settings. When there is norm mismatch between task vectors, applying DisTaC mitigates the interference between task vectors so that the normalized score is improved from the one of task arithmetic without DisTaC (e.g., RoBERTa-large: 58.1->80.5, +22.4 points in the normalized score). When the task vectors have low confidence, applying DisTaC improves the normalized score from the one without DisTaC (e.g., RoBERTa-large: 73.9->82.3, +8.4 points in the normalized score).
>
> These results demonstrate that (i) the two failure modes of norm disparity and low confidence we identify arise both in vision tasks and language tasks and (ii) DisTaC conditioning consistently improves merging outcome for CLIP/ViT and RoBERTa models. We believe these findings show modality generalizability of vision and language. Importantly, the recovery is stronger at larger scale (roberta-large), suggesting the approach remains useful as model capacity grows.
>
> We will incorporate these results and other merging methods (e.g., TIES-merging, Consensus merging, and TSVM) in the camera-ready version and revise the Introduction and Limitations to clarify the scope, generality, and modality positioning.
>
> [1] Wang, Alex, et al. "GLUE: A multi-task benchmark and analysis platform for natural language understanding." Proceedings of the 2018 EMNLP workshop BlackboxNLP: Analyzing and interpreting neural networks for NLP. 2018.

---

> ### Author Response · Authors · 2025-11-21
> **Response to reviewer XSii**
>
> # Regarding question 1
>
> We thank the reviewer for this important question regarding task interactions.
>
> DisTaC is explicitly designed as a pre-conditioner, and we view the resolution of inter-task interference (such as antagonistic or complementary relationships) as the primary role of the merging algorithm itself (e.g., TIES-Merging, Consensus Merging, TSVM).
>
> Crucially, DisTaC minimizes the risk of disrupting these inter-task relationships by preserving the direction of the original task vectors. Specifically, DisTaC imposes a strong $L_2$ regularization penalty to prevent deviation from the initial scaled task vector and typically concludes within a few training steps (e.g., 500 steps). This design ensures that the distillation process largely retains the original vector orientation, thereby preventing significant aggravation of interference or the destruction of complementary structures.

---

> > ### Comment · Reviewer_XSii · 2025-11-22
> > **Reply to Authors**
> >
> > Thank you for the author's reply, which partially addresses my concerns. I still have the following concerns:
> >
> > (1) Authors claim that DisTaC represents "the first systematic attempt to identify the specific failure modes (norm disparity and low confidence)," yet only a small set of conventional methods (i.e., Ties, TA, Consensus, and et al) are used to illustrate this problem. I suggest an essential extension of the evaluation to include a broader range of distinct and state-of-the-art merging methods, such as EMR-Merging, ISO-Merging, TSV-Merging, WUDI-Merging, and et al. Furthermore, a comprehensive analysis of the performance of these diverse methods under conditions of norm disparity and low confidence is crucial.
> >
> > (2) To provide more robust and generalizable support for the core research problem, the evaluation should be extended to LLMs and VLMs.

---

> > > ### Author Response · Authors · 2025-11-27
> > >
> > > Thank you for your feedback regarding the scope of our evaluation.
> > >
> > > As you suggested, we agree that a broader evaluation would further strengthen the validity of our arguments, and we conducted the following additional experiments.
> > >
> > > ## 1. Validation of a broader range of merging methods
> > >
> > > First, we would like to note that TSV-Merging (referred to as TSVM in our paper), which was included in your proposal, had already been evaluated at the time of submission and produced results consistent with our claims. We would like to emphasize that TSV-Merging was presented at CVPR 2025 and is one of the current state-of-the-art methods.
> > >
> > > In addition, we conducted evaluations of EMR-Merging, ISO-Merging, and WUDI-Merging as proposed. In the vision experiments in our paper, we evaluated both model sizes, ViT-B-32 and ViT-L-14, and present the results below.
> > >
> > > | ViT-B-32               |   Original  | Norm Mismatch | Low Confidence |
> > > |------------------------|-------------|---------------|----------------|
> > > | EMR-Merging            | 88.5 (98.4) |  80.0 (88.7)  |   39.2 (45.1)  |
> > > | EMR-Merging + DisTaC   |      -      |  **88.1 (97.3)**  |   **70.3 (79.2)**  |
> > > | ISO-Merging            | 81.0 (89.7) |  78.1 (86.2)  |   72.5 (81.1)  |
> > > | ISO-Merging + DisTaC   |      -      |  **80.3 (88.9)**  |   69.0 (78.1)  |
> > > | WUDI-Merging           | 85.5 (93.9) |  49.2 (52.6)  |   38.0 (40.8)  |
> > > | WUDI-Merging + DisTaC  |      -      |  **84.4 (93.2)**  |   **73.8 (83.3)**  |
> > >
> > > | ViT-L-14               |   Original  | Norm Mismatch | Low Confidence |
> > > |------------------------|-------------|---------------|----------------|
> > > | EMR-Merging            | 93.0 (99.6) |  87.6 (93.6)  |   27.4 (30.1)  |
> > > | EMR-Merging + DisTaC   |      -      |  **92.7 (99.0)**  |   **92.3 (98.1)**  |
> > > | ISO-Merging            | 90.4 (96.4) |  90.8 (96.9)  |   80.8 (86.0)  |
> > > | ISO-Merging + DisTaC   |      -      |  90.1 (96.1)  |   **86.1 (91.5)**  |
> > > | WUDI-Merging           | 91.7 (97.7) |  57.9 (60.8)  |   28.0 (29.2)  |
> > > | WUDI-Merging + DisTaC  |      -      |  **91.4 (97.5)**  |   **91.6 (97.3)**  |
> > >
> > > Excluding the two ISO-Merging cases, we observed substantial performance improvements with DisTaC in 10 out of 12 results overall. In particular, for WUDI-Merging on ViT-L-14, we observed extremely large improvements in absolute accuracy: **35.8%** for Norm Mismatch and **63.6%** for Low Confidence.
> > >
> > > We believe these results strengthen the evidence that the failure modes we identified manifest across a broader range of merging methods and can be mitigated by DisTaC.
> > >
> > > ## 2. Experiments with LLMs and VLMs
> > >
> > > First, we would like to state that for VLMs, our vision experiments use CLIP, which is a VLM. We hope this addresses your concern.
> > >
> > > For LLMs, we conducted evaluations using Llama‑2‑7B. As in our response to your weakness 2, we evaluated on the GLUE benchmark. The results are shown below.
> > >
> > > |                        |   Original  | Norm Mismatch | Low Confidence |
> > > |------------------------|-------------|---------------|----------------|
> > > | task addition          | 75.9 (91.7) |  55.3 (64.7)  |   75.7 (95.1)  |
> > > | task addition + DisTaC |      -      |  **75.0 (91.1)**  |   74.8 (**98.5**)  |
> > >
> > > Except for the absolute score in the Low Confidence setting, we observed performance improvements with DisTaC. In particular, for Norm Mismatch, DisTaC yielded nearly a **20%** increase in score, underscoring its significant effectiveness for LLMs as well.
> > >
> > > We have added the above results to Table 1 in the main text for merging methods, and to the Appendix for the LLM experiments.
> > >
> > > We hope the above addresses all of your concerns, and if so, we kindly request a reconsideration of the score.

---

> > > > ### Comment · Reviewer_XSii · 2025-11-28
> > > >
> > > > The authors' response has effectively addressed my previous concerns. The rebuttal demonstrates the prevalence of the two types of problems mentioned and verifies the general effectiveness of DisTaC. In light of this, I will raise my score to 6.

---

> > > > > ### Author Response · Authors · 2025-11-28
> > > > > **Reply to Reviewer XSii**
> > > > >
> > > > > Thank you for your thoughtful engagement with our submission.
> > > > >
> > > > > We are glad to hear that your concerns have been addressed, and we sincerely appreciate your decision to raise the score to 6.
> > > > > At present, the system still reflects 4, so we would be very grateful if you could kindly update it to 6 before the rebuttal period concludes.
> > > > >
> > > > > Thank you again for your careful consideration and for supporting the improvement of our work.

---

### Official Review · Reviewer_A7va · 2025-11-01

**Soundness:** 4
**Presentation:** 4
**Contribution:** 3
**Rating:** 6
**Confidence:** 4

**Summary:**

This paper investigates the robustness of multi-task model merging methods under realistic, non-idealized conditions. The authors identify and empirically validate two key factors that degrade the performance of existing merging techniques: (1) disparities in the norms of task vectors, often arising from different fine-tuning hyperparameters, and (2) low prediction confidence of the source models, which can result from calibration techniques like label smoothing. To address these issues, the paper proposes DisTaC, a pre-conditioning step that uses knowledge distillation on unlabeled data. DisTaC adjusts task vector norms and increases model confidence before the merge, thereby making state-of-the-art merging algorithms more robust. Experiments on vision tasks show that DisTaC significantly improves performance in these challenging scenarios, often recovering it to the level of an ideal merge.

**Strengths:**

1. The paper addresses a novel and highly relevant problem. While most research focuses on improving performance on idealized benchmarks, this work astutely investigates the robustness of merging methods in more practical, pessimistic settings, which is a critical step for real-world applicability.

2. The problem formulation is exceptionally clear and well-motivated. The failure modes are demonstrated with convincing empirical evidence (Figure 1), making the motivation for the proposed solution immediately apparent and easy to follow.

3. The paper provides valuable and actionable insights for practitioners. The discussions around "shrinking vs. stretching" task vectors and the trade-off between source model calibration and merge performance offer practical guidelines that extend beyond the method itself.

**Weaknesses:**

1. The empirical evaluation is limited in scope. All experiments are conducted on CLIP/ViT models for vision classification tasks. Given the current prominence of model merging in the context of Large Language Models (LLMs), the absence of any experiments on language tasks is a significant limitation and leaves the generalizability of the findings in question.

2. The investigation of failure modes, while insightful, feels somewhat narrow. The paper focuses on norm disparity and low confidence but does not explore other plausible sources of incompatibility, such as differences in optimizer states, dataset quality/size, or the specific architecture of parameter-efficient fine-tuning (e.g., LoRA rank).

3. The method introduces a dependency on unlabeled data from the task distribution. While the authors argue this is a reasonable requirement, it makes the method less self-contained than truly data-free merging techniques and may be a practical constraint in some scenarios.

4. Proposition 1 formalizes a basic geometric intuition about vector addition but offers little deep insight into the functional consequences within a neural network. The choice of cosine similarity as the primary metric is also debatable, as its connection to actual task performance is not explicitly established.

**Questions:**

Please see the weaknesses.

---

> ### Author Response · Authors · 2025-11-21
> **Response to reviewer A7va**
>
> # Regarding weakness 1
>
> We appreciate your concern that our evaluation initially focused on CLIP/ViT models for vision tasks and did not include language tasks. We conducted additional experiments on RoBERTa-base and RoBERTa-large across four tasks from GLUE benchmark [1] to test the same failure modes (norm disparity and low confidence) and the effectiveness of DisTaC.
>
> | Model: roberta-base      |   Original  | Norm Mismatch | Low Confidence |
> |--------------------------|-------------|---------------|----------------|
> | Task arithmetic          | 60.9 (73.5) |  56.8 (68.5)  |   61.3 (72.6)  |
> | Task arithmetic + DisTaC |      -      |  **59.9** (**71.7**)  |   **62.5** (**74.6**)  |
>
> | Model: roberta-large     |   Original  | Norm Mismatch | Low Confidence |
> |--------------------------|-------------|---------------|----------------|
> | Task arithmetic          | 68.3 (82.4) |  46.0 (58.1)  |   64.5 (73.9)  |
> | Task arithmetic + DisTaC |      -      |  **64.4** (**80.5**)  |   **70.0** (**82.3**)  |
>
> In the two tables above, we showed experimental results with RoBERTa-base and RoBERTa-large. Absolute score is displayed outside of parentheses, whereas normalized score appears in parentheses. The score in the table is based on the average of evaluation metrics defined in GLUE benchmark. Compared to the Original setting, the normalized score degrades under both Norm Mismatch and Low Confidence settings. When there is norm mismatch between task vectors, applying DisTaC mitigates the interference between task vectors so that the normalized score is improved from the one of task arithmetic without DisTaC (e.g., RoBERTa-large: 58.1->80.5, +22.4 points in the normalized score). When the task vectors have low confidence, applying DisTaC improves the normalized score from the one without DisTaC (e.g., RoBERTa-large: 73.9->82.3, +8.4 points in the normalized score).
>
> These results demonstrate that (i) the two failure modes of norm disparity and low confidence we identify arise both in vision tasks and language tasks and (ii) DisTaC conditioning consistently improves merging outcome for CLIP/ViT and RoBERTa models. We believe these findings show modality generalizability of vision and language. Importantly, the recovery is stronger at larger scale (roberta-large), suggesting the approach remains useful as model capacity grows.
>
> We will incorporate these results and other merging methods (e.g., TIES-merging, Consensus merging, and TSVM) in the camera-ready version and revise the Introduction and Limitations to clarify the scope, generality, and modality positioning.
>
> [1] Wang, Alex, et al. "GLUE: A multi-task benchmark and analysis platform for natural language understanding." Proceedings of the 2018 EMNLP workshop BlackboxNLP: Analyzing and interpreting neural networks for NLP. 2018.

---

> ### Author Response · Authors · 2025-11-21
> **Response to reviewer A7va**
>
> # Regarding weakness 2
>
> We thank the reviewer for this insightful comment and agree that factors such as optimizer states, dataset characteristics, and PEFT architectures are important dimensions in the broader landscape of model merging.
>
> First, we emphasize that our primary focus is on the inherent properties of task vectors, treating their generating root causes as secondary. Factors such as optimizer states and dataset quality correspond to these root causes; while they can certainly induce norm disparity or low confidence—for instance, different optimizers impose varying norm constraints [2][3]—they ultimately manifest as the weight-space symptoms we address. In such cases, these factors become detrimental to merging and necessitate intervention via DisTaC. However, given the vast number of possible setting combinations, an exhaustive investigation of every root cause is beyond our scope.
>
> Furthermore, given practical constraints where practitioners often lack access to original optimizer states or full training data, we prioritized failure modes diagnosable and fixable solely via model weights and unlabeled data.
>
> Regarding LoRA, we conducted experiments but found no clear results where the rank or its variance consistently explained merge performance; consequently, we concluded that rank properties do not constitute a distinct failure mode comparable to the issues identified in our study.
>
> [2] Bernstein, Jeremy, and Laker Newhouse. "Old optimizer, new norm: An anthology." arXiv preprint arXiv:2409.20325 (2024).
> [3] Chen, Lizhang, Jonathan Li, and Qiang Liu. "Muon Optimizes Under Spectral Norm Constraints." arXiv preprint arXiv:2506.15054 (2025).

---

> ### Author Response · Authors · 2025-11-21
> **Response to reviewer A7va**
>
> # Regarding weakness 3
> Thank you for carefully pointing out the practical concerns regarding the reliance on unlabeled data originating from task distribution.
> We acknowledge that DisTaC's reliance on unlabeled data may pose limitations in certain specialized domains, such as those with stringent security requirements or complex data distributions like a comprehensive set for LLMs.
>
> While a fully data-free approach to addressing all failure modes remains the ideal, we assert that DisTaC represents the first systematic attempt to identify the specific failure modes (norm disparity and low confidence) that critically undermine state-of-the-art merging methods and to propose a practical, effective remedy.
>
> Current state-of-the-art merging techniques, including TSVM and others, are largely data-free aside from coefficient tuning, and perform well under idealized benchmark conditions. Yet, they suffer significant performance degradation under the realistic failure modes we introduced. We contend that the use of unlabeled data in DisTaC is currently the most potent and realistic auxiliary means to restore robustness in these challenging scenarios.
>
> Furthermore, we demonstrated DisTaC's practical viability by confirming its strong robustness to variations in both data size and quality.
>
> We first tested DisTaC's performance by varying the number of unlabeled samples per class (100, 200, 300, 400, and 500). The table below shows the average relative test accuracy across all tasks, where $100\%$ represents the test accuracy achieved using the full unlabeled dataset (2490 samples per class on average) for DisTaC. For comparison, we included results for distillation starting directly from the pretrained model ("Distill-from-Pretrained").
>
> || 100  | 200  | 300  | 400  | 500  |
> |-----------|------|------|------|------|------|
> | [Norm Mismatch]        |   |   |   |   |   |
> | Distill-from-Pretrained  | 71.1 | 75.7 | 83.1 | 88.2 | 89.0 |
> | DisTaC   | **96.0** | **96.0** | **97.3** | **98.6** | **99.0** |
> | [Low Confidence]        |   |   |   |   |   |
> | Distill-from-Pretrained      | 70.1  | 73.8  | 81.2  | 84.6  | 87.6  |
> | DisTaC     | **83.9**  | **87.4**  | **90.5**  | **91.0**  | **95.0**  |
>
> The results demonstrate DisTaC's strong robustness to limited data:
> - DisTaC achieves over $90\%$ of the full-data test performance with just $\mathbf{300}$ samples per class in both failure modes, and maintains over $80\%$ performance even with $\mathbf{100}$ samples (reaching $96\%$ in the Norm Mismatch case).
> - Compared to distillation from the pretrained model, DisTaC exhibits superior robustness. This highlights the methodological benefit of initializing distillation from the already scaled task vector ($\theta_{pre} + \kappa_t \tau_t$).
>
> We next assessed robustness to degraded data quality by introducing dataset shift via Gaussian blur during distillation. This setup simulates real-world conditions like variations in weather or camera quality. The blur strength is controlled by the kernel size (fixed at 5) and the intensity range ($\sigma_{\min}, \sigma_{\max}$), where a larger $\sigma$ value indicates stronger corruption.
> The results below show the relative test accuracy against the performance achieved using clean data for distillation.
>
> |    | [0.1, 1] | [0.1, 2] | [1, 3] |
> |-----------|----------|----------|--------|
> | [Norm Mismatch]        |   |   |   |   |   |
> | Distill-from-Pretrained  | 98.1     | 95.7     | 90.7   |
> | DisTaC  | **100.4**    | **96.2**     | **91.7**   |
> | [Low Confidence]        |   |   |   |   |   |
> | Distill-from-Pretrained  | 98.1     | 97.3     | 94.7   |
> | DisTaC  | **99.6**    | **98.5**     | **99.9**   |
>
>
> The analysis confirms DisTaC's high robustness to quality degradation:
> - DisTaC consistently maintains performance, achieving over $90\%$ of the clean-data performance even under the most severe corruption ($\sigma_{\max}=3$).
> - In the challenging Low Confidence case, DisTaC maintains near-perfect accuracy (over $98.5\%$) regardless of corruption intensity.
> - DisTaC demonstrates superior robustness compared to the baseline, suggesting that utilizing the original fine-tuned model as the teacher effectively filters noise present in the unlabeled data.
>
> **This robust performance suggests that DisTaC remains effective even in real-world scenarios where sufficient high-quality unlabeled data may be difficult to gather.** We agree that achieving a complete data-free solution for these identified failure modes is extremely challenging and remains an important direction for future work.
>
> We have appended the above discussion to the paper’s appendix for completeness.

---

> ### Author Response · Authors · 2025-11-21
> **Response to reviewer A7va**
>
> # Regarding weakness 4
>
> We thank the reviewer for highlighting the need to bridge our geometric intuition with functional consequences. To explicitly establish the connection between cosine similarity and actual task performance, we analyze the merge within the Neural Tangent Kernel (NTK) regime. In this regime, the addition of two task vectors can be approximated linearly around the initialization $\theta_0$:
> $$f(x;\theta_0 + \tau_1 + \tau_2) \approx f(x;\theta_0) + (\tau_1 + \tau_2)^{\top}\nabla_\theta f(x;\theta_0)$$
> Under this linearization, the change in the model's functional behavior from pretrained model is given by $\Delta f(x) \approx (\tau_1 + \tau_2)^{\top}\nabla_\theta f(x;\theta_0) = \tau_{\text{merge}}^{\top}\nabla_\theta f(x;\theta_0)$. Since the Jacobian $\nabla_\theta f(x;\theta_0)$ is fixed at initialization, the functional shift is exclusively dictated by the merged vector $\tau_{\text{merge}}$. Consequently, the degree to which $\tau_{\text{merge}}$ is geometrically similar to $\tau_1$ and $\tau_2$ (as measured by cosine similarity) directly defines how closely the merged model's function approximates the respective fine-tuned models. Thus, the geometric argument in Proposition 1 effectively reduces to a discussion on task performance preservation. We have revised the relevant section of the manuscript to include this NTK-based explanation, thereby clarifying the functional implications of our geometric metrics.
>
> We have added this discussion to the explanation of Proposition 1 to make it easier to understand.

---

> ### Comment · Reviewer_A7va · 2025-11-27
>
> Thank you for the detailed response, which partially addresses my concern. I will maintain my positive score. Good luck.

---

> ### Author Response · Authors · 2025-11-27
>
> We thank the reviewer for the continued engagement. We suspect that your comment about our previous response only "partially" addressing the concern may stem from the fact that RoBERTa, while a language model, differs in scale from modern Large Language Models (LLMs).
> To fully resolve the concern regarding generalizability to LLMs, we have conducted additional experiments using Llama-2-7B on the GLUE benchmark.
>
> __Experiments on LLM (Llama-2-7B)__
> We evaluated the robustness of merging fine-tuned Llama-2-7B models under the identified failure modes. The results are summarized below:
>
>
> |                        |   Original  | Norm Mismatch | Low Confidence |
> |------------------------|-------------|---------------|----------------|
> | task addition          | 75.9 (91.7) |  55.3 (64.7)  |   75.7 (95.1)  |
> | task addition + DisTaC |      -      |  **75.0 (91.1)**  |   74.8 (**98.5**)  |
>
>
> Key Findings:
> - Norm Mismatch: DisTaC provides a substantial recovery, improving the absolute score by nearly 20 points (55.3 → 75.0) and recovering the normalized score to 91.1, which is comparable to the Original setting.
> - Low Confidence: DisTaC improves the normalized score (98.5 vs. 95.1), indicating that the merged model acts more similarly to the ideal single-task models, while maintaining comparable absolute performance to the baseline. These results on a 7B-parameter model confirm that DisTaC is highly effective for LLMs, scaling well beyond the RoBERTa and ViT architectures.
>
> We have added these Llama-2-7B results to the Appendix. We believe these additional experiments, combined with the RoBERTa results, provide comprehensive empirical evidence that the identified failure modes and the effectiveness of DisTaC are universal across Vision, Language, and their intersection (VLMs), regardless of model scale. We hope this fully addresses your concerns regarding generalizability and the scope of empirical evaluation. If so, we kindly request that you reconsider the score to reflect these extensive additions.

---

> ### Comment · Reviewer_A7va · 2025-11-28
>
> Thank you for the additional clarifications. I would like to briefly restate the primary considerations underlying my evaluation.
>
> A central advantage of model merging is that it typically operates directly in parameter space—through methods such as parameter addition, masking, or low-rank operations—incurring minimal computational overhead and not requiring gradient computation or backpropagation. By contrast, the proposed approach depends on knowledge distillation, gradient-based optimization, and the tuning of multiple hyperparameters. While the use of unlabeled data is commendable, this data requirement may still present practical limitations. Overall, these characteristics align the method more closely with traditional knowledge distillation techniques than with lightweight model merging.
>
> I appreciate the authors’ technical contributions and effort. However, the aforementioned concerns regarding elegance and practicality have influenced my decision not to assign a higher score.

---

> ### Author Response · Authors · 2025-11-28
> **Gratitude for the positive evaluation and reaffirming our main contribution**
>
> Thank you sincerely for the positive evaluation and for the constructive feedback throughout. Your comments helped us solidify the contribution of this work.
>
> The primary contribution of this paper is to systematically identify operational vulnerabilities in current model-merging methods that are obscured by idealized benchmarks, and to introduce DisTaC$-$a lightweight pre-conditioning approach$-$to mitigate them. As you noted, DisTaC requires distillation on unlabeled data, which introduces a small additional cost beyond traditional, data-free merging. **However, we wish to emphasize that without such auxiliary cost, existing merging methods suffer substantial performance degradation under the failure modes we highlight; this degradation is the more serious practical issue. In contrast, investing a minimal amount of distillation yields large and consistent gains, making DisTaC a rational and pragmatic choice in real deployments.** Moreover, prior work [1][2] has already recognized unlabeled-data-based, gradient-driven strategies as acceptable and effective tools within the model-merging landscape.
>
> Taken together, we believe this paper advances the practical reliability of model merging in real-world settings. We are grateful for your review and for the helpful comments, and again thank you for the positive assessment.
>
> [1] Yang, Enneng, et al. “AdaMerging: Adaptive Model Merging for Multi-Task Learning.” The Twelfth International Conference on Learning Representations (ICLR), 2024.
>
> [2] Yan, Kunda, et al. “CALM: Consensus-Aware Localized Merging for Multi-Task Learning.” Forty-second International Conference on Machine Learning (ICML), 2025.

---

### Comment · Area_Chair_CTiy · 2025-11-27

Dear Authors and Reviewers,

The discussion phase will end soon. If you want to further discuss comments and replies with each other, please post your thoughts by adding official comments.

Thanks for your efforts and contributions to ICLR 2026.

Best regards,

Your Area Chair

---

### Author Response · Authors · 2025-11-27
**Request for Re-evaluation**

Dear Reviewers and Area Chair,

We would like to express our sincere gratitude for your active engagement and constructive feedback throughout the discussion phase.

We are encouraged to hear that the reviewers have acknowledged our responses and revisions. Specifically, we appreciate the confirmation that our rebuttal has successfully addressed most of the concerns raised in the initial reviews.

Despite the consensus that the major technical concerns have been resolved and the manuscript has been significantly improved, we notice that the current ratings have not yet been updated to reflect this positive progress.

We respectfully ask you to reconsider your scores to align with your post-rebuttal assessment. If there are any remaining barriers preventing a score increase, please let us know immediately so we can address them before the discussion period closes.

We trust that our revisions and the resolved concerns will be fairly reflected in the final evaluation.

Best regards, The Authors

---

### Author Response · Authors · 2025-12-02
**Rebuttal Summary for AC**

Dear AC,

We sincerely appreciate the time and genuine effort dedicated to reviewing our work.

Below, we summarize the key points integrating the feedback from all four reviewers, along with our responses and additional experiments conducted during the rebuttal phase.

**1. Expanded Empirical Generalizability**

 Cross-Modality: We confirmed the reproducibility of failure modes and the efficacy of DisTaC not only in CLIP/ViT (VLM) but also in RoBERTa and Llama-2-7B. Notably, for Llama-2-7B under Norm Mismatch, DisTaC recovered performance by approximately +20 absolute points, and normalized metrics improved under Low Confidence scenarios.

Cross-Method: In addition to standard evaluations (TIES/Consensus/TSVM), we tested EMR/ISO/WUDI-Merging. We observed significant degradation under failure modes across these methods and confirmed substantial recovery with DisTaC (e.g., for WUDI-Merging w/ ViT-L-14, Norm Mismatch improved from 57.9% to 91.4%, and Low Confidence from 28.0% to 91.6% in absolute accuracy).

**2. Data Requirements & Robustness**

Robustness: We confirmed high robustness even with small-scale or low-quality unlabeled data. For instance, using only 300 samples per class achieved >90% of ideal performance; performance remained >90% even under strong Gaussian blur, and >98.5% in Low Confidence settings across all blur intensities.

Refined Claims: While we originally described unlabeled data as "readily available," we have revised the manuscript to acknowledge domain dependencies for academic rigor. However, our results demonstrate that DisTaC remains effective even when data quantity or quality is suboptimal.

**3. Computational Cost & Practicality**

Efficiency: For ViT-B-32, the process requires only 500 steps ($\approx$3.2 seconds on 2×A100) with a peak memory of $\approx$7.1GB.

Selective Application: We emphasized that DisTaC is intended for selective application to specific task vectors based on diagnosis, rather than all vectors. This presents a practical rationale: standard merging degrades significantly under failure modes, whereas DisTaC offers a consistent, low-cost remedy.

**4. Theoretical Integrity**

Link to Functionality: Addressing concerns that Proposition 1 (Norm Mismatch) was limited to parameter space, we utilized NTK approximation to explicitly link norm divergence to functional degradation, proving that parameter discrepancies lead to performance drops.

Interpretation of Sec 6.1: We clarified that the improved single-task performance observed via task vector shrinking can be theoretically interpreted as a regularization effect equivalent to early stopping.

**Conclusion**

This work identifies hidden vulnerabilities in real-world model merging and provides a practical, low-cost solution that ensures stable recovery. Through extensive additional experiments and manuscript revisions, we have reinforced the generalizability, validity, and operational guidelines of our approach.

**Following the rebuttal, Reviewer XSii indicated their intention to raise their score from 4 to 6. Consequently, all reviewers now effectively align with a positive score (6), showing a unanimous consensus towards acceptance.**

We once again thank all reviewers and the AC for their constructive engagement to make our contribution more solid.

Best regards,

The Authors

---

### Meta-Review · Area_Chair_3yZu · 2026-01-01

**Summary:**

This paper systematically investigates the robustness of existing model merging methods under realistic, non-idealized settings and identifies two critical yet previously underexplored failure modes: task vector norm mismatch and low confidence of source models. To address these issues, the authors propose DisTaC, a knowledge-distillation-based pre-conditioning approach that adjusts task vector norms and improves model confidence prior to merging. Beyond the initial vision-focused experiments, the authors substantially strengthened the paper during the rebuttal by adding evaluations on NLP models (RoBERTa), large-scale LLMs (Llama-2-7B), and a broader set of state-of-the-art merging methods. Overall, the work provides a coherent diagnosis–treatment pipeline for robust model merging and presents convincing empirical evidence across modalities and model scales.

A key strength of this work is the practical relevance of the problem it tackles. Rather than optimizing performance on idealized benchmarks, the paper exposes systematic vulnerabilities of existing merging techniques that are highly likely to occur in real deployments. The identified failure modes are clearly motivated, easy to diagnose, and empirically well supported. DisTaC itself is conceptually simple yet effective, and its role as a lightweight, modular pre-conditioner makes it compatible with a wide range of existing merging algorithms. The experimental evaluation is thorough, especially after rebuttal: the authors demonstrate robustness across vision, language, and large-scale models, include multiple merging paradigms, and provide careful analyses of data requirements, data quality sensitivity, and computational overhead. The additional theoretical discussion linking geometric properties of task vectors to functional behavior further improves the clarity of the contribution.

The main limitation is that DisTaC departs from the fully data-free and purely parameter-space philosophy that motivates many model merging methods, as it relies on unlabeled data and gradient-based distillation. While the authors convincingly show that the computational cost is small and that the method can be applied selectively, this requirement may still be restrictive in data-scarce or security-sensitive settings. In addition, the theoretical analysis remains largely explanatory rather than predictive, and does not yet offer strong guarantees beyond the studied failure modes.

Overall, this is a solid and well-executed paper that makes a meaningful contribution by identifying important, previously overlooked robustness issues in model merging and offering a practical, effective remedy. The authors responded thoroughly and constructively to reviewer concerns, significantly strengthening the empirical scope and clarity of the work. Despite some loss of elegance compared to strictly data-free merging, the gains in robustness and real-world applicability are substantial. I am inclined toward acceptance.

**Reviewer Concerns:**

Most major reviewer concerns were addressed in the rebuttal. In particular, limitations on empirical scope and generalizability were convincingly resolved through additional experiments on NLP models (RoBERTa), large-scale LLMs (Llama-2-7B), and a broader set of merging methods, which reviewers explicitly acknowledged by maintaining or raising their scores. Practical concerns about reliance on unlabeled data and computational overhead were also largely addressed via quantitative runtime analysis, robustness studies under limited or noisy data, and clarification that DisTaC is a selective, diagnostic-driven pre-conditioning step rather than a mandatory component.

Remaining concerns are relatively minor. DisTaC still departs from fully data-free, purely parameter-space merging, which may be undesirable in some settings, and the added theoretical analysis remains mainly explanatory rather than providing strong formal guarantees. Overall, no significant technical issues remain unresolved.

**Reviewer Scores:**

Reviewer A7va: Likely unchanged at 6. The reviewer explicitly stated they would maintain their positive score after the rebuttal, and their remaining concerns were more about elegance and philosophy rather than correctness or impact.

Reviewer XSii: Increased from 4 to 6. The reviewer explicitly acknowledged that the rebuttal addressed their concerns and stated their intention to raise the score to 6.

Reviewer P2H4: Likely unchanged at 6. The reviewer indicated that most concerns were addressed, maintained a positive score, and only suggested minor wording refinements.

Reviewer bqXy: Likely unchanged at 6. The reviewer’s main concerns about modality generalization and data sensitivity were directly addressed in the rebuttal, and they expressed a clearly positive overall assessment.

Overall, if all reviewers had fully participated throughout the discussion, the final consensus would be uniformly positive, centered around scores of 6.

---

### Decision · Program_Chairs · 2026-01-26

Accept (Poster)